# Balancing Preservation and Modification:
# A Region and Semantic-Aware Metric for Instruction-Based Image Editing

Zhuoying Li [1] [*]   Zhu Xu [1] [*]   Yuxin Peng [1]   Yang Liu [1]

## Abstract

Instruction-based image editing, which aims to modify the image faithfully towards instruction while preserving irrelevant content unchanged, has made advanced progresses. However, there still lacks a comprehensive metric for assessing the editing quality. Existing metrics either require high costs concerning human evaluation, which hinders large-scale evaluation, or adapt from other tasks and lose specified concerns, failing to comprehensively evaluate the modification of instruction and the preservation of irrelevant regions, resulting in biased evaluation. To tackle it, we introduce a new metric **B**alancing **P**reservation **M**odification (**BPM**), that tailored for instruction-based image editing by explicitly disentangling the image into editing-relevant and irrelevant regions for specific consideration. We first identify and locate editing-relevant regions, followed by a two-tier process to assess editing quality: *Region-Aware Judge* evaluates whether the position and size of the edited region align with instruction, and *Semantic-Aware Judge* further assesses the instruction content compliance within editing-relevant regions as well as content preservation within irrelevant regions, yielding comprehensive and interpretable quality assessment. Moreover, the editing-relevant region localization in BPM can be integrated into image editing approaches to improve the editing quality, manifesting its wild application. We verify the effectiveness of BPM metric on comprehensive instruction-editing data, and the results show that we yield the highest alignment with human evaluation compared to existing metrics, indicating efficacy. The code is available at https://joyli-x.github.io/BPM/.

[*]Equal contribution [1]Wangxuan Institute of Computer Technology, Peking University. Correspondence to: Yang Liu <yangliu@pku.edu.cn>.

*Proceedings of the 42nd International Conference on Machine Learning*, Vancouver, Canada. PMLR 267, 2025. Copyright 2025 by the author(s).

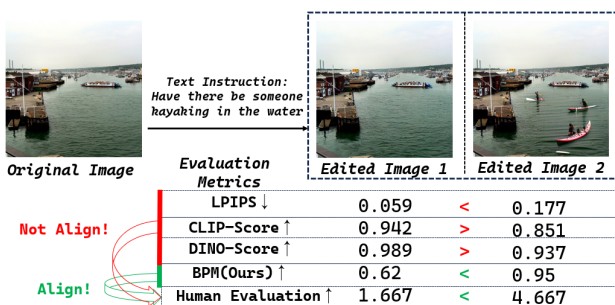

(a) Results of evaluation metrics for identical sample.

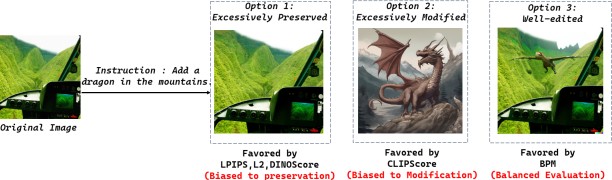

(b) Visualized Example of Ground Truth Test.

Figure 1: (a) The scores rated by existing metrics LPIPS, CLIP-Score and DINOScore all witness a contradict trends with human evaluation, while our proposed BPM aligns with human evaluation. (b) Previous metrics favor excessively preserved or modified result, while our BPM favor the well-edited image.

## 1. Introduction

Instruction-based image editing (Hertz et al., 2022; Kawar et al., 2022; Brooks et al., 2022; Han et al., 2024; Wang et al., 2024) has gained increasing attention, which aims to modify the given image via text instructions, requiring the edited image faithfully align with the modification requirements while preserving the unrelated parts of the origin image. Though a great success, the field still lacks a sufficiently standardized metric for comprehensively assessing the editing quality. Ideally, we would expect the most authoritative evaluations to come from human assessments. However, the costs and overhead associated with human evaluation are too high to scale, hindering its practical value in large-scale evaluations. In response to it, a series of existing works (Hertz et al., 2023; Basu et al., 2023; Gal et al., 2022; Kim & Ye, 2021; Brooks et al., 2022; Ruiz et al., 2023; Kocasari et al., 2022) have proposed automated evaluation metrics.

One line of metrics evaluate the alignment between edited image and text instruction, such as CLIPScore (Hessel et al., 2021). Other metrics, like DINOScore (Liu et al., 2023), LPIPS (Zhang et al., 2018) and CLIP-I assesses the similarity between original images and edited images. But these metrics often perform contradict human evaluations in cases, leading their results deemed unreliable. As shown in Fig. 1(a), two edited images are rated by different existing metrics, the right image successfully incorporates "human kayaking" to fulfill instruction requirements, gaining higher human evaluation score. However, all existing metrics show an opposite conclusion, highlighting their untrustworthiness. We emphasize that the flaws in current automatic metrics arise from the following reasons: (1) These metrics are **not specifically tailored** for this task, typically originating from task evaluation for image generation. They **fail to simultaneously utilize** <**Origin Image, Edited Image, Instruction** > elements of instruction-based editing task, ignoring crucial information: overlooking of instruction fails to judge the modification quality of target editing element, while overlooking of origin image results in poor judgment of the preservation quality of regions should not be modified. (2) These metrics **evaluate the entire image as a whole, without accounting for the fact that some regions of the image need to be preserved, while others require modification**. As a result, they fall short of assessing the different requirements for editing process. To verify it, we conduct a ground truth test[1]: for an input image and editing instruction, apart from the well-edited result (which serves as ground truth), we introduce two types of images generated by holistic operations performed on the entire image for comparison. One is directly adding noise to the original image, serving as an excessively preserved global edit. The other is directly generated by a Text-to-Image(Rombach et al., 2021) model based on the instruction, serving as an excessively modified global edit. One visualized example is shown in Fig. 1(b). Existing metrics favor excessively modified or preserved results that acquired from global editing on entire image, manifesting their evaluation are biased without separate consideration for different image regions. Evaluating image as a whole also lead to the inability to assess whether the changes of edited region properties, such as size and position, follows the instruction. The above reasons reveal the urgent need to develop a more comprehensive metric to evaluate instruction-based image editing.

To this end, we propose **B**alancing **P**reservation **M**odification (BPM) tailored for instruction-based image editing. The core idea is to explicitly disentangle the image region that should be modified or preserved, separately evaluating the semantic instruction compliance for the modified region and the content preservation for the irrelevant region, thus endowing the evaluation process

with comprehensiveness and interpretability. However, given the diversity nature of instructions, accurately localizing the editing regions is non-trivial: for complex instruction involving object size change and spatial relation requirement, the extent of scaling ("make the apple bigger") and displacement ("add an apple to the right of banana") during the editing process makes editing region highly flexible, necessitates referring to both original and edited image to determine it. To enable such accurate localization, we parse the instruction by a large language model(LLM) to determine the source and target object during editing, and utilize detection and segmentation tools to locate the editing regions (both in original and edited images). Then, we propose a two-tier evaluation process: *Region-Aware Judge* verify whether the size and position of edited region aligns with instruction, ensuring the overall editing region is roughly correct; Subsequently, *Semantic-Aware Judge* is adopted to conduct more fine-grained evaluation from semantic perspective: for the localized edited region, we retain the directional CLIP similarity to assess whether the modification faithfully complies with the instruction; for irrelevant regions, we adopt L2-distance to calculate whether the original content has been successfully preserved. Such hierarchical evaluation facilitates a comprehensive and nuanced assessment of all requirements for editing task. Experimentally, our BPM witnesses remarkable alignment improvement with human evaluation on diverse instruction-based editing data, indicating that BPM can provide more trustworthy evaluation results in comparison with previous metrics. Furthermore, we find the modification region localization process within BPM can be seamlessly integrated into existing image editing approaches, guiding the editing more focused on target editing region while reducing unnecessary modifications to irrelevant regions, yielding more satisfactory results.

The major contributions are summarized as follows. (1) We delve into the drawback analysis of existing metrics for instruction-based image editing; (2) We introduce a novel metric BPM for instruction-based image editing, which evaluates editing quality from the region and semantic perspective with explicit disentangled separate consideration for editing relevant and irrelevant region, yielding comprehensive and interpretable evaluation score; (3) Through extensive experiments, BPM demonstrates a high correlation with human evaluation and outperforms previous metrics, indicating its efficacy; (4) We utilize region localization process within BPM to provide a training-free enhancement for existing image editing approaches, yielding more accurate edited results with fewer irrelevant modifications.

## 2. Related Works

**Conventional Image Editing Quality Evaluation Metrics**

---

[1]The full experiment and detailed analysis is in Sec. 4.

Existing metrics evaluate the editing quality from two different perspectives: the first line of metrics, such as LPIPS (Zhang et al., 2018), DINOScore (Liu et al., 2023) and L2 distance, measures the similarity between the edited and original image, but they overlook to evaluate the modification quality for the given instruction; the other line of metrics, such as CLIPScore (Gal et al., 2021), evaluates the success of an edit by assessing the semantic alignment between the edited image and the instruction. However, such metrics overlook that instruction-based editing requires both image content that should be modified and preserved. All above metrics fail to utilize the "origin image, edited image, instruction" information for comprehensive evaluation. Besides, we emphasize that simply combining the two lines of metrics is insufficient: they all serve the entire image as a whole for evaluation, which is against the fact that different image regions encompass different requirements, i.e., preservation or modification. In contrast to their approach, our BPM not only explicitly separates the image into relevant and irrelevant regions for editing, allowing for distinct evaluation, but also incorporates all three elements of the editing process. This enables the assessment of both modification and preservation, as well as the accuracy of the edited regions and content, resulting in a more interpretable evaluation score.

**VLM-based Image Quality Evaluation Metrics**

Considering the great reasoning ability and visual understanding ability of Vision language models (VLMs), several prior works adopt VLMs as automatic evaluators for image quality. (Ku et al., 2024; Lu et al., 2024; Li et al., 2024; Xu et al., 2024; Hui et al., 2024) proposes to utilize vision language models to automatically evaluate text-to-image generation task by assessing image-text alignment. HQ-Edit (Hui et al., 2024) utilizes GPT-4o (OpenAI, 2024) to evaluate the image editing from coherence and alignment perspectives. However, even the state-of-the-art VLMs like GPT-4o(OpenAI, 2024), are reported to show a decent capability in such image quality assessment concerning image-text alignment, and mistakes are often witnessed(OpenAI, 2023) during such evaluation process, hindering the trustworthy of VLM-based approaches. Instead of relying on VLMs for the whole evaluation process, we only adopt them to parse the instruction to locate the edit-relevant regions, thus alleviating the above mentioned problem.

## 3. Method

### 3.1. Overview

This section presents BPM , a novel evaluation metric that enables accurate and comprehensive editing quality assessment. The whole process is shown in Alg. 1. Considering the overlook of simultaneous utilization of origin image

---

**Algorithm 1** BPM Evaluation Process

---

1: **Input:** Original Image $I_{origin}$, Edited Image $I_{edit}$, text editing instruction $T_{edit}$

2: **Functions:** large language model LLM to analyze $T_{edit}$, function $F(\cdot, \cdot, \cdot, \cdot, \cdot, \cdot)$ calculates the modification and preservation score for edited image. $D(\cdot, \cdot)$ detects and segments objects of specific category and return corresponding bounding box and mask. $f_{size}$ and $f_{position}$ verify the attribute alignment in terms of edited object's size and position.

3: **Output:** Region-Aware Evaluation Score $S_{region}$ and Semantic-Aware Score $S_{semantic}$.

4: $o_1, o_2, pos_{st}, size_{st} \leftarrow$ LLM($T_{edit}$) {parse the instruction to acquire the source and target object $o_1, o_2$, and the size state change $size_{st}$(larger, smaller, unchanged) and location change $pos_{st}$(left, right, up, down, unchanged) during the editing process. }

5: $M_{origin}, B_{origin} \leftarrow$ D($I_{origin}, o_1$)

6: $M_{edit}, B_{edit} \leftarrow$ D($I_{edit}, o_2$)

7: $S_{position} \leftarrow f_{position}(B_{origin}, B_{edit}, pos_{st}, I_{edit})$

8: $S_{size} \leftarrow f_{size}(M_{origin}, M_{edit}, size_{st})$

9: $S_{modify}, S_{preserve} \leftarrow$ F($I_{origin}, I_{edit}, B_{origin}, B_{edit}, M_{origin}, M_{edit}$)

10: $S_{region} \leftarrow S_{position} + S_{size}$

11: $S_{semantic} \leftarrow S_{modify} + S_{preserve}$

12: $BPM \leftarrow \alpha * S_{semantic} + (1 - \alpha) * S_{region}$

---

$I_{origin}$, edited image $I_{edit}$ and instruction $T_{edit}$ in existing metrics results in biased evaluation, we incorporate all three elements for calculation. We first explicitly decompose both $I_{origin}$ and $I_{edit}$ into editing relevant and irrelevant regions. It is accomplished by first adopting a Large Language Model (Line 4) to parse instruction to acquire the source and target object $o_1, o_2$ involving the editing process, then utilizing detection and segmentation tools $D(\cdot)$ to separately locate the source object $o_1$ and target object $o_2$ in original image $I_{origin}$ and edited image $I_{edit}$, acquiring bounding box $B_{origin/edit}$ and mask $M_{origin/edit}$ (Line 5-6) to disentangle the whole image. We also guide the LLM to output the required edited object size and position changing state $size_{st}, pos_{st}$ during editing, which are utilized to conduct following *Region-Aware Judge*, which aims to verify the editing process aligns with the instruction in terms of the size and location changing. We design specific functions $f_{psosition}(\cdot)$ and $f_{size}(\cdot)$ acquiring scores for region size and position alignment (Line 7-8), termed $S_{position}$ and $S_{size}$, which together form Region-aware score $S_{region}$. To enable more fine-grained *Semantic-Aware Judge*, we adopt function $F(\cdot)$ to evaluate the semantical instruction compliance within editing region ($S_{modify}$) as well as content preservation for irrelevant region ($S_{preserve}$) (Line 8), which together as Semantic-Aware score $S_{semantic}$. Finally, we obtain the BPM score by adding $S_{semantic}$ and $S_{region}$

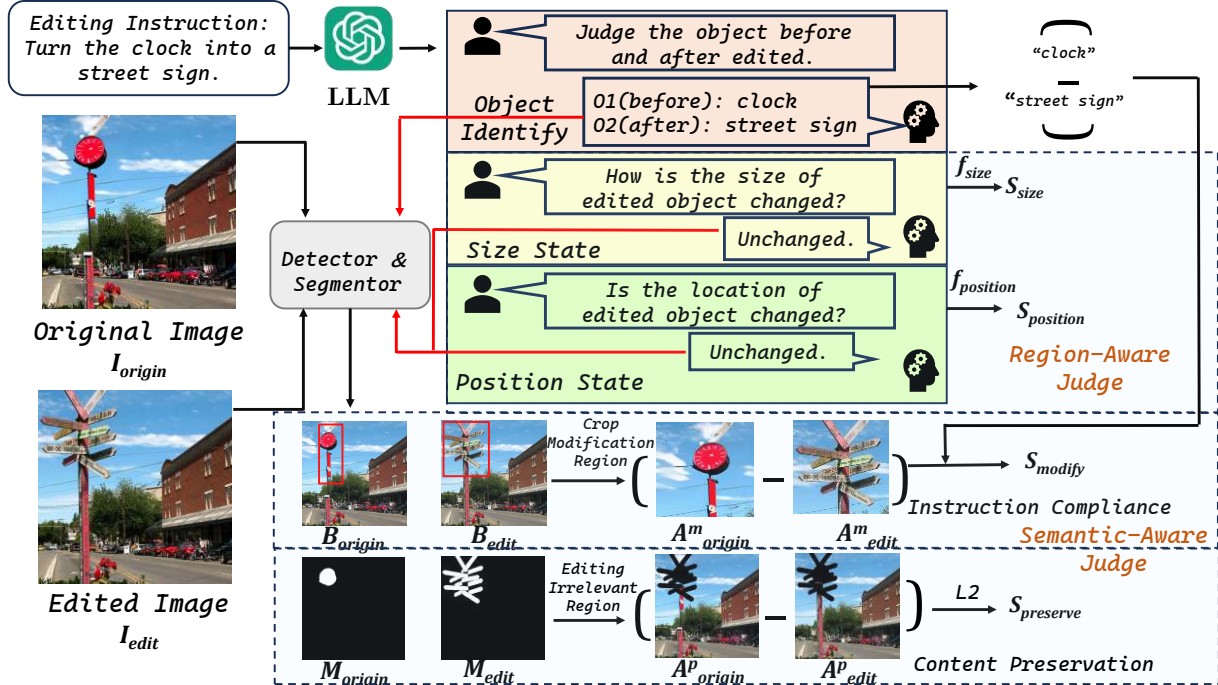

Figure 2: **Overall Pipeline of BPM .** Firstly, LLM is utilized to parse and analyze editing instruction, generates responses to identify the source and target object, as well as the object size and position changing state requirement during editing. Then we conduct *Region-Aware Judge* to verify editing follows the instruction for region size and position, yielding region-aware score $S_{region}$. For *Semantic-Aware Judge*, we utilize detection and segmentation tools to locate and segment the edited object, subsequently apart the origin and edited images into edited regions and irrelevant regions, separately evaluating the semantical instruction compliance in edited regions and content preservation in irrelevant regions, yielding semantic-aware score $S_{semantic}$.

with different weights (Line 12).

In the following, we separately introduce how we parse the instruction and localize the modified regions in Sec. 3.2, the *Region-Aware Judge* (object position and size alignment verification) in Sec. 3.3, the *Semantic-Aware Judge* (semantical instruction compliance within modified regions and content preservation for irrelevant regions) in Sec. 3.4. Finally, we delve into an attempt to utilize the editing region localization process within BPM to improve the editing quality of image editing methods in Sec. 3.5.

### 3.2. Instruction Parsing and Editing Region Localization

We employ LLM to parse the editing instruction $T_{edit}$ based on three criteria, involving the editing object identification, object repositioning and size modifications judgement. One example is shown in Fig. 2. The model is required to first decide the exact object category involved in the editing process, outputting *clock* as source object $o_1$ and *street sign* as target object $o_2$. What's more, LLM is also guided to determine whether the editing is required to affect the size of the corresponding edited object (outputting "larger," "unchanged," or "smaller", termed $size_{st}$) and how the object's

position undergoes displacement (outputting "left", "right", "up", "down" or "unchanged", termed $pos_{st}$). In the example, the instruction requires no specific position and size change, and $pos_{st}$ and $size_{st}$ is "unchanged". Then for editing region localization, considering that the diverse editing instructions necessitates simultaneous consideration of both the original image $I_{origin}$ and edited image $I_{edit}$ to accurately determine the modified regions, we employed detection and segmentation models to identify the position of $o_1$ and $o_2$ in $I_{origin}$ and $I_{edit}$, represented as bounding boxes $B_{origin/edit}$ and instance masks $M_{origin/edit}$, respectively. By incorporating the modified region within original image $I_{origin}$ as well as edited image $I_{edit}$, we can accurately acquire the modified region within the editing process. Specifically, for the case of "add object", the original object $o_1$ is set as "None", and the corresponding mask $M_{origin}$ and bounding box $B_{origin}$ are set identical as $M_{edit}/B_{edit}$ in default. Similarly, in the "remove object" case, the edited object $o_2$ is set to "None," and $M_{edit}/B_{edit}$ are initialized to be identical to $M_{origin}/B_{origin}$.

### 3.3. Region-Aware Judge

After acquiring the masks $M_{origin}, M_{edit}$ and bounding boxes $B_{origin}, B_{edit}$ for original object $o_1$ and edited object

$o_2$, we verify the size and position changing is aligned with text instruction $T_{edit}$, acquiring the region-aware evaluation score. We provide two rule-based functions, termed size judgment ($f_{size}$ in Alg. 1) and position judgment ($f_{position}$ in Alg. 1) respectively, as shown below.

**Position Alignment:** The pipeline is shown in Fig. 3(a). We utilize LLM to acquire the position changing state $pos_{st}$ required in instruction, which denotes spatial relation between edited object and reference object. Specifically, the edited and reference object is identical for instructions without explicit reference object, such as "Move the apple to the right." With the bounding boxes of edited and reference object, we employ three criteria to determine whether the editing meet the requirement: (1)*Position Assessment:* The center coordinates of bounding boxes are used for direct position assessment; (2)*Direction Verification:* The difference between the x- and y-coordinates is required to check the existence of unwanted edits in unrelated directions; (3)*Saliency Check:* A certain spatial distance between the edited object and the reference object (expressed as bounding box IOU less than a threshold) is further required to check the editing saliency. If all three criteria are satisfied, we consider the editing comply with instruction and score $S_{position}$ as 1; otherwise, we score $S_{position}$ as 0. Particularly, for "unchanged" instructions, we only evaluate it through *Saliency Check* by requiring the bounding box IOU is greater than a threshold.

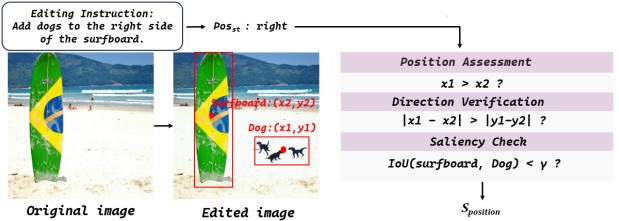

(a) Position Alignment verification.

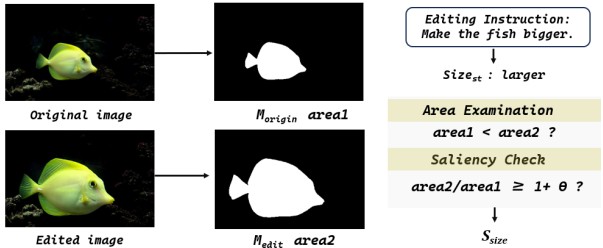

(b) Size Alignment verification.

Figure 3: **Evaluation functions for region-aware judge.**

**Size Alignment:** The pipeline is shown in Fig. 3(b). The area of $M_{origin}$ and $M_{edit}$ represents the original and edited object size, and we evaluate whether the size change conforms to required size change state $size_{st}$ with following criteria: (1)*Area Examination:* Size comparison of $M_{origin}$ and $M_{edit}$ is directly adopted to examine the requirements.

(2)*Saliency Check:* The ratio of the front and rear area needs to be greater than (for "larger" as $size_{st}$) or less than (for "smaller" as $size_{st}$) a certain threshold to check the changing saliency. According to above criteria, we yield score $S_{size}$ as 1(conformed) or 0(disobey). Instructions with "unchanged" $size_{st}$ is evaluated by *Saliency Check*, restricting the ratio of areas to be around 1 within a certain range.

---

**Algorithm 2** Semantic-Aware Judge Process
___

1: **Input:** Original Image $I_{origin}$, Edited Image $I_{edit}$, mask $M_{origin}$ $M_{edit}$, bbox $B_{origin}$, $B_{edit}$.
2: **Functions:** $C_B$ crops image regions following given bbox, $CLIP_I$ and $CLIP_T$ are CLIP image encoder and text encoder. $Norm$ is min-max normalization.
3: **Output:** modification score $S_{modify}$ and preservation score $S_{preserve}$.
4: $A_{origin}^m \leftarrow C_B(I_{origin}, B_{origin})$
5: $A_{edit}^m \leftarrow C_B(I_{edit}, B_{edit})$
6: $A_{origin}^p \leftarrow C_M(I_{origin}, M_{origin})$
7: $A_{edit}^p \leftarrow C_M(I_{edit}, M_{edit})$
8: $A_{origin}^p \leftarrow (1 - M_{origin} \cup M_{edit}) \odot I_{origin}$
9: $A_{edit}^p \leftarrow (1 - M_{origin} \cup M_{edit}) \odot I_{edit}$
10: $S_{modify} \leftarrow \cos\_sim\{CLIP_I(A_{edit}^m)\text{-}CLIP_I(A_{origin}^m),$ $CLIP_T(o_2)\text{-}CLIP_T(o_1)\}$
11: $S_{preserve} \leftarrow 1 - L2(A_{origin}^p, A_{edit}^p)$
12: $S_{semantic} \leftarrow Norm(S_{preserve}) + Norm(S_{modify})$

---

### 3.4. Semantic-Aware Judge

Furthermore, to conduct more fine-grained evaluation from a semantic perspective, we leverage the obtained bounding box and mask information to disentangle the origin and edited image $I_{origin}$ and $I_{edit}$ for separately evaluating the quality of semantical instruction compliance and content preservation. The process is shown in Alg. 2.

**Semantic Instruction Compliance:** the obtained bounding boxes $B_{origin}, B_{edit}$ are utilized to isolate the edited object's region, thereby excluding interference from unmodified areas and enhancing the quality assessment for modification, yielding cropped edited regions $A_{origin}^m$ for $I_{origin}$ and $A_{edit}^m$ for $I_{edit}$ (Line 4-5). Then, to measure the semantic consistency between the editing results and instructional specifications, following CLIP Directional Similarity (Kim et al., 2022), we calculate the cosine similarity between the difference of $A_{origin}^m$, $A_{edit}^m$ and corresponding text embedding of $o_1$, $o_2$ in CLIP space as follows:

$$S_{modify} = \text{cs}\Big(CLIP_I(A_{origin}^m) - CLIP_I(A_{edit}^m), \\ CLIP_T(o_1) - CLIP_T(o_2)\Big) \quad (1)$$

where $\text{cs}(\mathbf{a}, \mathbf{b}) = \frac{\mathbf{a} \cdot \mathbf{b}}{\|\mathbf{a}\|\|\mathbf{b}\|}$ denotes cosine similarity, $CLIP_I(\cdot)$ and $CLIP_T(\cdot)$ is the CLIP image encoder and text

encoder. The rationale behind such directional similarity calculation is that by computing the difference in features before and after editing, we can not only assess whether the features requiring modification align with the instruction, but also implicitly verify that other attributes unrelated to the editing process are successfully retained, thus providing a more comprehensive reflection of correspondence between editing and instruction. Higher $S_{semantic}$ indicates better semantic instruction compliance.

**Irrelevant Content Preservation:** we utilize masks to exclude modified regions to only retain the preserved parts. Considering the potential object displacement during editing, we utilize the union of $M_{origin}$ and $M_{edit}$ to exclude all areas intended for editing, which is formulated as:

$$A^p_{origin} = [1 - M_{origin} \cup M_{edit}] \odot I_{origin} \qquad (2)$$

$$A^p_{edit} = [1 - M_{origin} \cup M_{edit}] \odot I_{edit} \qquad (3)$$

where $\cup$ is the union operation between masks, and $\odot$ is the element-wise matrix product. Considering the potential object displacement during editing, we utilize the union of $M_{origin}$ and $M_{edit}$ to exclude all areas intended for editing, and subsequently obtain the preserved portion for original image $A^p_{origin}$ and edited image $A^p_{edit}$. Then, we measure the L2-distance between $A^p_{origin}$ and $A^p_{edit}$, subsequently acquiring preservation score $S_{preserve}$ = 1 - L2($A^p_{origin}, A^p_{edit}$). Higher $S_{preserve}$ denotes better content preservation.

Ultimately, we acquire the semantic aware score $S_{semantic}$ by normalizing and combining $S_{modify}$ and $S_{preserve}$, which accurately assess the instruction compliance in edited region, as well as the content preservation in irrelevant regions.

### 3.5. Application for Editing Quality Enhancement

Furthermore, considering that the process in Sec. 3.2 enables accurate editing region localization, we delve into an attempt to utilize the editing region located as the guidance for image editing model, leading the model to focus attention on the regions that need editing, thereby reducing modifications to unrelated areas and achieving results that better align with the editing instructions. Specifically, we first acquire the edited image using original editing model, and adopt the region localization process in BPM to acquire mask of edited area in both $I_{origin}$ and $I_{edit}$: $M_{all} = M_{origin} \cup M_{edit}$. Then in the second-round editing quality enhancement process, $M_{all}$ serves as explicit guidance through classifier-free guidance (Ho & Salimans, 2022) with following formulation,

$$\begin{aligned} \epsilon_\theta(z_t, t, I_{origin}, T_{edit}) = &\epsilon_\theta(z_t, t, \varnothing, \varnothing) + \\ &s_I * (\epsilon_\theta(z_t, t, I_{origin}, \varnothing) - \epsilon_\theta(z_t, t, \varnothing, \varnothing)) + \\ &s_T * (\epsilon_\theta(z_t, t, I_{origin}, T_{edit}) - \epsilon_\theta(z_t, t, I_{origin}, \varnothing)) \odot M_{all} \end{aligned} \qquad (4)$$

where the $z_t$ is the denoised latent vector during editing process, $t$ is the timestep, $s_I$ and $s_T$ are the guidance scale for input image $I_{origin}$ and instruction $T_{edit}$, respectively, $\odot$ is hadamard product operation. By adopting such class-free guidance, the instruction guidance is restricted within the region that is relevant to the editing process, thus reducing unnecessary modifications in other regions, subsequently yielding more satisfactory editing results.

## 4. Experiement

### 4.1. Implementation Details

We employ a pre-trained CLIP-ViT 14/B(Radford et al., 2021) model for directional similarity calculation. For the LLM for instruction parsing, we select gemma-2-9b-it-SimPO(Meng et al., 2024) in seek of trade-off between cost and performance. For detector and segmentor, we use Grounding DINO (Liu et al., 2023) and Grounded SAM (Ren et al., 2024). For scaling weight $\alpha$ for $S_{semantic}$, we set it to 0.7 (i.e. BPM = $0.7 * S_{semantic} + 0.3 * S_{region}$. More results with different LLMs, further details on prompting the source and target descriptions and the impact of different $\alpha$ are referred to the appendix.

**Evaluation setting**. We conduct ***Human Alignment Test*** and ***Ground Truth Test***. ***Human Alignment Test*** assess the correlation between metric evaluation and human judgment. To implement it, we first conduct user study, asking users to rate the editing images from 1(worst) to 5(best).[2] For two images $I^1_{edit}$ and $I^2_{edit}$ edited by two different editing models with identical input, we examine the consistency in the ranking of their human scores $H_1, H_2$ and metric scores $M_1, M_2$. Human ranking $P_h$ is calculated by comparing $H_1$ and $H_2$, where $P_h = \mathbb{1}_{H_1 > H_2}$. Similarly, we calculate the metric ranking as $P_m = \mathbb{1}_{M_1 > M_2}$. Finally we calculate the alignment ratio of $P_m$ and $P_h$ across all the samples as *Alignment Score*$= \frac{1}{N} \sum_{k=1}^N \mathbb{1}_{P^k_m = P^k_h}$, where N is the sample number. Higer *Alignment Score* indicates better alignment between human and metric evaluation, and we report it for each pair editing model for comprehensive comparison. ***Ground Truth test*** assesses whether the metrics correctly identify the well-edited image within a triplet of images categorized as excessively modified, excessively preserved, ground truth(well edited). The ground truth are manually filtered and selected as well-edited image, while the excessively modified images are directly generated by a text-to-image model (Rombach et al., 2021) according to the instruction, the excessively preserved images are generated by adding Gaussian noise to the original image. We report the proportion of favoring for each metric across these three types of images, with the total sum equal to 1. A higher proportion of favoring for ground truth images indicates

---

[2]The user study details are provided in Appendix.

**Table 1: Human alignment test on BPM and existing image editing metrics for local edits.** We report the alignment of preference between metric and human evaluation for paired editing images. Bold and underline represent the highest and second-best performance.

| | MGIE vs FT$_{IP2P}$ ↑ | MGIE vs IP2P↑ | IP2P vs FT$_{IP2P}$ ↑ | DaLLE-2 vs IP2P↑ | DALLE-2 vs FT$_{IP2P}$ ↑ | DALLE-2 vs MGIE↑ | Average ↑ |
|---|---|---|---|---|---|---|---|
| *Preservation Metrics* | | | | | | | |
| **L2** | 0.651 | 0.614 | 0.724 | 0.762 | 0.659 | 0.801 | 0.702 |
| **LPIPS** | **0.687** | 0.602 | 0.665 | 0.731 | 0.626 | 0.791 | 0.683 |
| **CLIP-I** | 0.578 | 0.574 | 0.571 | 0.705 | 0.643 | 0.796 | 0.644 |
| **DINOScore** | 0.608 | 0.597 | 0.606 | 0.746 | 0.692 | 0.796 | 0.674 |
| *Modification Metrics* | | | | | | | |
| **CLIPScore** | 0.530 | 0.477 | 0.494 | 0.394 | 0.429 | 0.387 | 0.452 |
| *VLM-based Metrics* | | | | | | | |
| **GPT-4o** | 0.681 | **0.676** | 0.712 | 0.860 | 0.780 | 0.817 | 0.754 |
| *Balanced Metrics* | | | | | | | |
| **CLIPScore +CLIP-I** | 0.648 | 0.667 | 0.607 | 0.691 | 0.573 | 0.660 | 0.641 |
| **BPM** | 0.663 | 0.659 | **0.818** | **0.943** | **0.802** | **0.911** | **0.799** |

**Table 2: Human alignment test on BPM and existing image editing metrics for global edits.**

| | PIE vs HQ_EDIT ↑ | PIE vs FT$_{IP2P}$ ↑ | HQ_EDIT vs FT$_{IP2P}$ ↑ | FT$_{IP2P}$ vs IP2P ↑ | IP2P vs HQ_EDIT ↑ | IP2P vs PIE↑ | Average ↑ |
|---|---|---|---|---|---|---|---|
| *Preservation Metrics* | | | | | | | |
| **L2** | 0.897 | 0.889 | 0.333 | 0.400 | 0.520 | 0.925 | 0.661 |
| **LPIPS** | **1.000** | **0.917** | 0.375 | 0.467 | 0.480 | 0.900 | 0.690 |
| **CLIP-I** | 0.966 | 0.861 | 0.417 | 0.600 | 0.520 | 0.850 | 0.702 |
| **DINOScore** | 0.966 | 0.833 | 0.250 | 0.533 | 0.520 | 0.900 | 0.667 |
| *Modification Metrics* | | | | | | | |
| **CLIPScore** | 0.897 | 0.833 | **0.917** | 0.667 | 0.760 | 0.825 | 0.817 |
| *VLM-based Metrics* | | | | | | | |
| **GPT-4o** | 0.759 | 0.722 | 0.583 | 0.733 | 0.760 | 0.825 | 0.730 |
| *Balanced Metrics* | | | | | | | |
| **CLIPScore +CLIP-I** | **1.000** | **0.917** | 0.458 | **0.750** | 0.600 | **1.000** | 0.787 |
| **BPM** | 0.897 | 0.889 | 0.750 | **0.750** | **0.840** | 0.850 | **0.829** |

**Table 3: The inference speed of metrics.**

| Metric | BPM (Ours) | CLIP-T | CLIP-I | LPIPS | DINO | L2 | GPT-4o |
|---|---|---|---|---|---|---|---|
| Speed (seconds per image) | 1.500 | 0.225 | 0.350 | 0.067 | 0.050 | 0.028 | 11.400 |

**Table 5: Effectiveness verification for component score of** $S_{semantic}$. $S_{preserve}$ and $S_{modify}$ separately targets for content preservation and instruction compliance quality evaluation, we compare them with corresponding human evaluation.

| | | IP2P vs FT$_{IP2P}$ ↑ | DaLLE-2 vs IP2P↑ | DALLE-2 vs FT$_{IP2P}$ ↑ |
|---|---|---|---|---|
| preservation | L2 | 0.816 | 0.893 | 0.743 |
| | $S_{preserve}$ | **0.829** | **0.893** | **0.757** |
| modification | CLIPScore | 0.624 | 0.484 | 0.479 |
| | $S_{modify}$ | **0.659** | **0.67** | **0.606** |

**Table 4: Ground Truth test.** We report the winning rate of three type of images for each editing metric. "*EP*" represents "Excessively Preserved", "*EM*" represents "Excessively Modified", "*GT*" represents "Ground Truth". Each row sums to 1, indicating how many portion of images the metric at that row favors.

| Metrics | EP | EM | GT |
|---|---|---|---|
| **L2** | 0.83 | 0 | 0.17 |
| **LPIPS** | 1 | 0 | 0 |
| **CLIP-I** | 0.87 | 0 | 0.13 |
| **DINOScore** | 0.98 | 0 | 0.02 |
| **CLIPScore** | 0.04 | 0.83 | 0.13 |
| **CLIPScore +CLIP-I** | 0.42 | 0.14 | 0.44 |
| **GPT-4o** | 0.08 | 0.14 | 0.78 |
| **BPM-overall** | 0.12 | 0.01 | 0.87 |

**Table 6: Effectiveness verification for** $S_{position}$ **and** $S_{size}$**.** We conduct user study to verify the alignment between human and metric judgment towards editing region size and position change.

| | MGIE ↑ | IP2P ↑ | DaLLE-2 ↑ | FT$_{IP2P}$ ↑ |
|---|---|---|---|---|
| $S_{size}$ | 0.89 | 0.89 | 0.93 | 0.92 |
| $S_{position}$ | 0.71 | 0.69 | 0.89 | 0.75 |

better evaluation for that metric.

For evaluation setting that compares the correlation between human scores and metric scores, we explain in the supplementary materials the reasons why this type of experiment is not recommended in this task, as well as the relevant experimental results.

**Compared Metrics.** We compare BPM with three lines of existing automatic metrics: (1) *Preservation Metrics* focuses on the similarity of original and edited image, including DINOscore (Liu et al., 2023), LPIPS (Johnson et al., 2016), CLIP-I and L2 distance. (2) *Modification Metrics* evaluate alignment between edited image and text instruction, where CLIPScore (Hessel et al., 2021) is the only one. We also incorporate the trade-off between CLIPScore and other metrics, like CLIPScore+CLIP-I, for more comprehensive comparison. (3) *VLM-based Metrics* utilize VLM to automatically evaluate the editing quality, we select most advanced VLM GPT-4o (OpenAI, 2024) with the evaluation prompts in HQ-Edit (Hui et al., 2024) for comparison.

**Evaluation Datasets**. For local editing, we sample image-instruction pairs from MagicBrush (Zhang et al., 2024), which covers six mainstream instruction editing types: adding objects, replacing objects, changing actions, changing colors, removing objects and changing patterns. Then we employ four advanced instruction-based image editing models IP2P (Brooks et al., 2022), MGIE (Fu et al., 2023), DALLE-2 (Ramesh et al., 2022), and $FT_{IP2P}$ (Zhang et al., 2024) to generate edited images. For global editing, we sample image-instruction pairs from the global edits subset of PIE-Bench (Ju et al., 2023). In addition to the baseline of PIE-bench itself, we also used HQ-Edit (Hui et al., 2024), IP2P (Brooks et al., 2022), and $FT_{IP2P}$ (Zhang et al., 2024) to generate edited images. Finally, we get 960 entries with human score annotation.

## 4.2. Comparison with other evaluation metrics

***Human Alignment Test*** result is shown in Table. 1 (local edits) and Table. 2 (global edits). For local edits, we can conclude that: (1) for each editing model pair comparison, overall score ($S_{semantic} + S_{region}$) of BPM yield the highest alignment with human evaluation compared to existing editing metrics that evaluating editing quality from preservation and modification perspective in most cases(Line 1-5), indicating BPM can serve as a more trustworthy and authentic metric for image editing evaluation; (2) we also merge the score of CLIPScore and CLIP-I to serve as a stronger comparison (Line 7) with both consideration of preservation and modification, and BPM still outperforms it, indicating simply summarizing preservation and modification metrics is insufficient, as they overlook to separate consider the requirements for region relevant or irrelevant to editing process, and BPM resolves it by explicitly disentangling the image for separate evaluation, yielding more satisfactory results. (3) BPM outperform the VLM-based evaluation powered by GPT-4o(OpenAI, 2024) (Line 6), indicating that directly adopting VLM for automatic evaluation exist untrustworthiness, and we mitigate such problem to enable more authentic evaluation. Besides, as shown in Table. 3. our BPM shows much faster inference speed than GPT-4o, validating our practicality. For global edits, we can conclude that: (1) Our BPM still yields the highest human alignment compared to other metrics, showing its effectiveness and wild application scope. (2) It is noted that the ClipScore shows a significant improvement compared to local edits. This may be because, in global edits, a larger number of pixels are modified, and the shortcomings of ClipScore in terms of overlooking original image are less pronounced compared to local edits.

***Ground Truth Test*** performance is shown in Tab. 4, each row represents the metric favoring proportion for three types of images and sum up to 1, higher proportion for *GT* indicates less biased evaluation results. We can conclude that:

(1) preservation metrics (L2, LPIPS, CLIP-I, DINOScore) prefer excessively-preserved results, while modification metrics (CLIPScore) prefer excessively-modified results, indicating all these metrics exist biased evaluation, leading their evaluation deemed unreliable. (2) Our BPM -overall favors the ground truth images, i.e., the well-edited ones with 87% ratio and surpasses GPT-4o results, which is credit to our explicit disentanglement of editing relevant and irrelevant regions for separate consideration, alleviating the biased towards the preservation or modification.

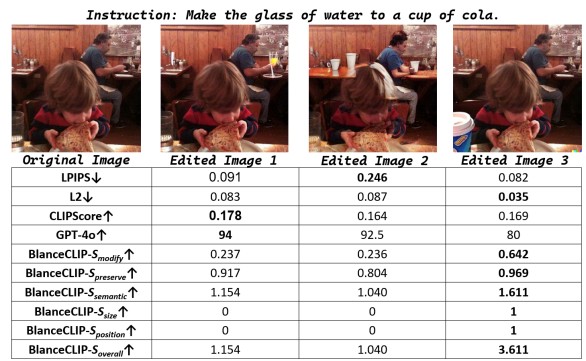

Figure 4: **Visualized example of evaluation metrics comparison.**

## 4.3. Ablation Study

Several ablation studies are conducted to separately verify the effectiveness of (1) each component score in BPM . (2) LLM instruction parsing. (3) our directional similarity.

**Effectiveness of Component Scores:** To verify the editing quality from each perspective respectively, we further ask human annotators to separately rate the edited image quality for preservation, modification, size, and position, and report the componential evaluation alignment. (1) for preservation and modification, we compared our scores with L2 and CLIPScore to validate the necessity of our explicit image region decomposition, the results are in Tab. 5, $S_{preserve}$ and $S_{modify}$ consistently outperform L2 and CLIPScore, highlighting that decomposing the image into editing relevant or irrelevant regions for separate consideration is crucial to acquire accurate assessment for modification and preservation, as it removes the interference of regions that owns opposite requirements; (2) for size and position, we report the alignment between human and our metric evaluation for size and position change, as shown in Tab. 6. Our $S_{position}$ and $S_{size}$ both yield relatively high alignment with human evaluation, indicating our design is effective in judging the position and size changing during editing.

**Effectiveness of LLM Instruction Parsing:** we validate it from two aspects, the accuracy of editing region localization as well as the size and position changing judgment

Table 7: **Ablation for our directional Similarity.** "CS" denotes vanilla CLIPScore for the cropped edited object region, "DS" denotes our directional similarity. We report the alignment score between human and metric evaluation.

| | IP2P vs FT$_{IP2P}$ ↑ | DaLLE-2 vs IP2P↑ | DALLE-2 vs FT$_{IP2P}$ ↑ | DALLE-2 vs MGIE↑ |
|---|---|---|---|---|
| CS | 0.635 | 0.619 | 0.479 | 0.536 |
| DS | **0.659** | **0.670** | **0.606** | **0.583** |

alignment with instruction. To verify it, we randomly select 100 data samples along with their edited region mask $M_{origin}$, $M_{edit}$ and required position and size changing state $Pos_{st}$, $Size_{st}$, then manually judge (1) whether the $M_{origin}$ and $M_{edit}$ accurately locate the editing region. (2) whether the $Pos_{st}$ and $Size_{st}$ express the requirement in instruction. Finally, we yield 97% accurate editing region localization and 99% correct position and size changing requirement expression, validating the trustworthiness of our LLM instruction parsing, subsequently facilitating the ultimate editing evaluation.

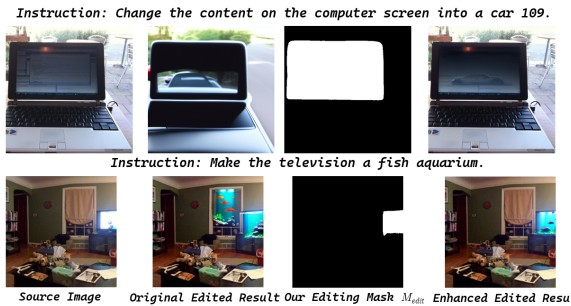

Figure 5: **Visualized example of image editing mask guidance.**

**Effectiveness of Directional Similarity:** we compared our directional similarity with original CLIPScore, i.e., the image-text alignment of edited object. From the results in Table. 7 we can see that our directional similarity yields higher alignment with human evaluation, validating that directional similarity indeed implicitly incorporates the verification of editing-irrelevant attributes maintenance, thus providing more reliable reflection of editing quality.

### 4.4. Application as Image Editing Guidance

We adopt three advanced image editing models MGIE(Fu et al., 2023), FT$_{IP2P}$(Zhang et al., 2024) and InstrucPix2Pix(Brooks et al., 2022), and separately implement our enhancement in Sec. 3.5 for them. The performance is shown in Tab. 8. For both preservation and modification, steady alignment improvement are observed across all three editing models, indicating that by adopting our mask guidance, the model can focus on correct editing region, thus reducing unnecessary modifications in irrelevant regions to enhance the preservation performance, as well as harvesting more high-quality modification within editing regions to

improve the modification performance. We also validate the quality enhancement through user study, asking users to select better edited image from original and our enhanced edited images. Our method achieved 91% of the samples being rated as better or equally good as the original, refer to Appendix for details.

Table 8: **Results for Editing Quality Enhancement.**

| Methods | Preservation | | Modification | |
|---|---|---|---|---|
| | Origin | Ours | Origin | Ours |
| IP2P | 0.792 | **0.927** | 0.318 | **0.334** |
| FT_IP2P | 0.907 | **0.922** | 0.361 | **0.364** |
| MGIE | 0.876 | **0.923** | 0.327 | **0.341** |

### 4.5. Qualitative Results

We present qualitative results to separately verify (1) BPM can comprehensive evaluating the edited images with trustworthy scoring. In Fig. 4, we compare three editing images for identical input image and instruction. The edited image 3 owns accurate editing within the region of source object "glass", as well as good content preservation in other irrelevant regions. Our BPM favors it in terms of preservation, modification as well as overall quality, highlighting the alignment of our evaluation towards human judgment; (2) the edited region localization process can enhance editing performance. In Fig. 5, our editing region mask $M_{edit}$ accurately locate the region should be edited, guiding the model to focus within it, reducing unnecessary modifications in other regions, thus harvesting more high-quality editing results. More qualitative results are in Appendix.

## 5. Conclusion

In this paper, we propose a novel metric BPM tailored for instruction-based imaged editing, which explicitly disentangle the image into editing-relevant and irrelevant regions for separate evaluation according to their requirements. Two-tier evaluations are conducted from region and semantic perspectives, yielding comprehensive editing quality assessment. What's more, the editing region localization in can be integrated into image-editing models to enhance the edited image quality, indicating its wild application. Experiments upon editing data with diverse instructions verify the effectiveness of BPM serving as a trustworthy evaluation metric.

## Impact Statement

Image editing quality evaluation has broad impacts: while facilitating community development, it also raises privacy and misinformation concerns that require careful attention to ensure responsible deployment and mitigate risks.

# Acknowledgements

This work was supported by the grants from National Natural Science Foundation of China (62372014,62525201, 62132001, 62432001) and Beijing Natural Science Foundation (4252040, L247006).

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

In the appendix, we first provide more information concerning method details, LLM prompts and user studies in Sec. A. Then we provide extra quantitative results for (1) impact of different scaling weight in Sec. B.1; (2) results of correlation between metrics and human evaluation score in Sec. B.2. (3) ablation study about the choice of LLM in Sec. B.3; (4) validation of LLM parsing result in Sec. B.4 (5) user study of masks generated by different pipeline in Sec. B.5 (6) more comprehensive component score ablation in Sec. B.6. (7) user study results for masking guidance enhancement in Sec. B.7. Finally, we provide qualitative results to validate (1)the quality of our editing region mask(in Sec. C.1); (2) the effectiveness of our BPM evaluation.(in Sec. C.2) (3) the effectiveness of our mask guided enhancement.(in Sec. C.3).

## A. Method Details

### A.1. LLM instruction parsing Prompts

The prompts for LLM instruction parsing is shown in Figure. 6. Our prompts include three parts, separately targets for object identify, size state and position state judgement.

### A.2. User Study for Human Alignment Test

In this part, we provide details about the user study regarding the degree to which the metrics align with human ratings.

**User Study Implementation Details:** We invited 3 participants to rate the quality of the provided editing images through a questionnaire. We randomly select 100 image-instruction pairs from MagicBrush(Zhang et al., 2024), and adopt IP2P(Brooks et al., 2022),MGIE(Fu et al., 2023), DALLE-2(Ramesh et al., 2022), and $FT_{IP2P}$(Zhang et al., 2024) to generate edited images. Participants were instructed to rate images from 5 different perspectives, each score from 1 to 5, representing bad quality to perfect quality. Evaluation criteria are (1) modification quality, which assesses whether the editing accurately comply the instruction; (2) preservation quality: which assesses whether regions should not be edited is successfully preserved; (3) size accuracy, which assesses whether the editing region/object size change follows the instruction; (4) position accuracy, which assesses whether the editing region/object position change follows the instruction; (5)overall accuracy, which assesses the overall editing quality by considering above perspectives together. Detailed user study instruction and standards are shown in Fig. 7. We ultimately yield 960 user responses, which is comprehensive enough to conduct the human alignment test. We also provide one screen shot for our user study page in Fig. 8.

### A.3. User Study for Mask Guided Enhancement Quality

To analyze whether the editing performance have improved after adding mask guidance, we conducted preference tests with three human evaluators comparing images produced by the original models and those generated by models with mask guidance. We randomly selected 100 original image-instruction pairs. For each pair, we randomly chose one model from IP2P (Brooks et al., 2022), MGIE (Fu et al., 2023), and $FT_{IP2P}$ (Zhang et al., 2024) to perform image editing, and compared it with the version that included mask guidance. The users were asked to choose the better image between the edited image produced by the original model and the edited image produced by the mask-guided model, or to consider them equivalent. The results from the three evaluators were then aggregated by majority vote; if all three chose differently, that sample was discarded. Detailed user study instruction and standards are shown in Fig. 9. We also provide one screen shot for our user study page in Fig. 10.

### A.4. $f_{size}$ and $f_{position}$ corner case discussion:

There exists some corner cases for size and position judge process: (1)For the add type instruction, where the source object in the original image is "None", $f_{size}$ directly assigns full score to size state. (2)For the remove type instruction, where the target object in the edited image is "None", $f_{position}$ directly assigns full score to position state.

## B. More Experimental Results

### B.1. Ablation on Different Balance Factors

We conduct experiments with different weight factor $\alpha$ between $S_{semantic}$ and $S_{region}$, i.e., $BPM = \alpha * S_{semantic} + (1 - \alpha) * S_{region}$, the results are shown in Table. 9. Among all combinations, $\alpha = 0.7$ reaches the best human evaluation alignment, verifying that higher yet moderate semantic scale would yield better evaluation results.

| | |
|---|---|
| **Object identify** | Given an instruction for image editing, output the parts of the original image that should be modified and the modified parts in the edited image compared to the original image. If there are no parts that should be modified, output None. Don't output the reasons.
Here are some examples:
1. Instruction: "Replace the bike with bear." Output: Original image: bike. Edited image: bear.
2. Instruction: "Add a bear." Output: Original image: None. Edited image: bear.
3. Instruction: "Remove the bike." Output: Original image: bike. Edited image: None.
Here is the instruction: |
| **Size state** | Given an instruction for image editing, output what size changes of the objects are involved, such as "larger", "smaller", "unchanged". Don't output the reasons.
Here are some examples:
1. Instruction: "give the bear bigger claws." Output: larger.
2. Instruction: "Make the hydrant blue." Output: unchanged.
3. Instruction: "Make the vase smaller." Output: smaller.
Here is the instruction: |
| **Position state** | Given an instruction for image editing and edited area in the original and edited image, output the positional relationship between the edited area in the edited image and the edited area in the original image. The optional positional relationships are: "left", "right", "up", "down", "inside", "unchanged", "near".
Here are some examples:
1. Instruction: "put a car on the screen of the laptop". Edited area in original image: screen of the laptop. Edited area in edited image: car. Output: inside.
2. Instruction: "put a bird on top of the tower". Edited area in original image: tower. Edited area in edited image: bird. Output: up.
3. Instruction: "replace the cat with a dog". Edited area in original image: cat. Edited area in edited image: dog. Output: unchanged.
Here is the instruction: |

Figure 6: **Prompts for parsing instruction.**

Table 9: **Ablation on different balance factors between** $S_{semantic}$ **and** $S_{region}$**.** Experiment conducted on a 100-entries subset.

| $\alpha$ | 0 | 0.1 | 0.2 | 0.3 | 0.4 | 0.5 | 0.6 | **0.7** | 0.8 | 0.9 | 1 |
|---|---|---|---|---|---|---|---|---|---|---|---|
| Human Alignment Score | 0.302 | 0.423 | 0.598 | 0.659 | 0.738 | 0.793 | 0.819 | **0.822** | 0.776 | 0.769 | 0.761 |

### B.2. Correlation between metrics and human evaluation score

We report metrics and human scores correlation for each single model in Tab. 10. Though our BPM yield highest correlation in this format as well, such calculation is questionable as output from distinct instruction lacks comparability, and all metrics show performance not high.

The rationale behind our pair model preference calculation is a) *Reduce Subjectivity influence*: Scores among users exhibit variance of 1.06(out of 0-5 range), thus using absolute values for single model correlation calculation may introduce bias. In contrast, employing relative preferences, by introducing a reference, reduces subjectivity in user study to enhance the credibility of the results. b) *Reduce cross-sample bias*: Note our metric evaluates which model performs better for the same instruction, rather than comparing scores across different instructions, as we believe outputs from distinct instructions lack absolute comparability, as shown in Figure. 11. By pair-wise comparison on identical instructions, we enable more accurate model-to-model comparisons explicitly than correlating scores from a single model's diverse outputs.

### B.3. LLM Selection Influence

We alter different LLM to parse the instruction, and the performance comparison is shown in Table. 11. From the result we can conclude that for the task of parsing instructions, current advanced LLM show similar alignment performance between metric and human evaluation. Specifically, Gemma-9B can achieve performance on par with GPT-4o.

# Instruction

Please assess whether the edited image meets the editing requirements. The assessment is divided into five dimensions: overall editing quality, quality of the edited area, consistency of the non-edited area, whether the size of the edited object meets the editing instructions, and whether the position of the edited object is correct.

The assessment criteria are as follows:

- Overall editing quality: Evaluate the overall quality of the generated image, considering the realism of the edited image, whether the edits align with the editing instructions and appear natural, and whether the non-edited areas remain consistent with the original image.
- Quality of the edited area: Assess whether the modified areas required by the instructions meet the textual editing instructions and appear natural.
- Consistency of the non-edited area: Evaluate whether the areas that should not be edited remain consistent with the original image.
- Size: Whether the size of the edited object meets the editing instructions, For example, "make the fire hydrant blue" means that the size of the fire hydrant should remain unchanged; while "make the vase bigger" means that the vase should increase in size.
- Position: Whether the position of the edited object is correct, For example, "make the hydrant blue," the position of the hydrant should remain unchanged; "put an apple on the table", the apple should be on top of the table.

The scoring range is an integer from 1 to 5:

- 1 point: Very Poor
- 2 points: Poor
- 3 points: Average
- 4 points: Good
- 5 points: Very Good

Figure 7: **The instructions for user study.**

**Editing instruction:**

Make the hydrant blue.

Figure 8: **Screen-shot of our user study page.**

# Instruction

Please judge which of the two edited images is better edited. The assessment is divided into five dimensions: overall editing quality, quality of the edited area, consistency of the non-edited area, whether the size of the edited object meets the editing instructions, and whether the position of the edited object is correct.

The assessment criteria are as follows:

- Overall editing quality: Evaluate the overall quality of the generated image, considering the realism of the edited image, whether the edits align with the editing instructions and appear natural, and whether the non-edited areas remain consistent with the original image.
- Quality of the edited area: Assess whether the modified areas required by the instructions meet the textual editing instructions and appear natural.
- Consistency of the non-edited area: Evaluate whether the areas that should not be edited remain consistent with the original image.
- Size: Whether the size of the edited object meets the editing instructions, For example, "make the fire hydrant blue" means that the size of the fire hydrant should remain unchanged; while "make the vase bigger" means that the vase should increase in size.
- Position: Whether the position of the edited object is correct, For example, "make the hydrant blue," the position of the hydrant should remain unchanged; "put an apple on the table", the apple should be on top of the table.

The scoring range is an integer from -1 to 1:

- 1 point: the left edited image is better than the right one
- 0 points: equal
- -1 points: the right edited image is better than the left one

Figure 9: **The instructions for user study of preference test.**

Table 10: **Correlation between metrics and human evaluation score on each single editing model.**

|  | MGIE | FT_IP2P | IP2P | DALLE2 | GenArtist | ACE | Average |
|---|---|---|---|---|---|---|---|
| CLIP-T | 0.114 | -0.028 | -0.146 | 0.079 | 0.280 | 0.075 | 0.062 |
| CLIP-I | 0.207 | -0.069 | 0.201 | 0.018 | 0.632 | 0.260 | 0.208 |
| DINO-Score | 0.315 | 0.039 | 0.252 | 0.115 | 0.695 | 0.299 | 0.285 |
| LPIPS | 0.264 | -0.034 | 0.276 | 0.153 | 0.735 | 0.198 | 0.265 |
| L2 | 0.210 | 0.082 | 0.156 | 0.256 | 0.628 | 0.151 | 0.247 |
| GPT-4o | **0.512** | **0.464** | **0.491** | 0.338 | 0.425 | **0.430** | 0.443 |
| BPM( Ours) | 0.474 | 0.374 | **0.452** | **0.488** | **0.537** | 0.324 | **0.444** |

## B.4. Validation of LLM Parsing Result

We randomly sampled 100 examples, used Gemma-9b for parsing, and manually evaluated the accuracy. As shown in Table. 12, gemma's parsing accuracy has reached over 97%. Specifically, the failure cases of the parsing of object identify are "Could it be eggs?" and "It could be just french fries." The errors occurred because the instructions contained "it" and since the LLM did not have information from the original image, it was unable to determine what "it" referred to, leading to mistakes. For the parsing of positions, the errors were due to the LLM's difficulty in judging other spatial relationships that were not included in the types we provided, such as "among."

## B.5. Mask Comparison

To quantitatively confirm the quality of our masks (i.e., edited regions), we randomly sampled 100 entries and asked 3 human evaluators to score the masks generated by different methods on a scale of 1(worse) to 5(best). These methods include the bounding boxes generated by GPT-4o, the masks produced by DiffEdit(Couairon et al., 2022), and the masks generated by our pipeline. As shown in Table. 13, our average mask quality score is 4.01, which is significantly higher than the other two methods. In addition to quantitative experiment, some visual results are presented in Figure. 12.

**Editing instruction:**

Make the hydrant blue.

**Original image:**          **Generated image 1:**          **Generated image 2:**

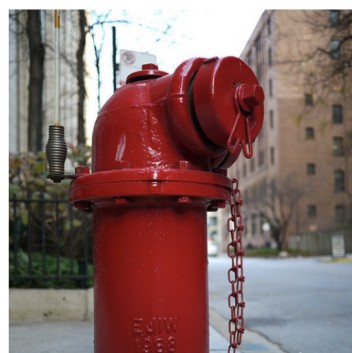 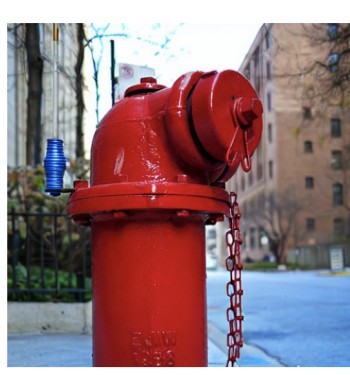 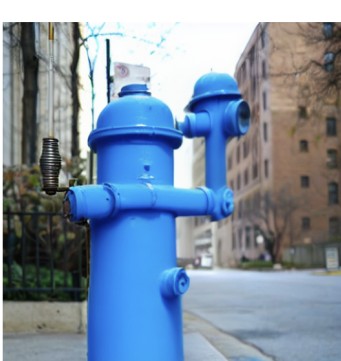

Quality of the edited area:

Consistency of the non-edited area:

Size:

Position:

Overall editing quality:

Figure 10: **Screen-shot of our user study page for preference test.** Users were asked to evaluate whether the left edited image is better than the right one from five dimensions.

### B.6. Additional Component Score Verification

Due to limited space, the full component score verification is shown in Table. 14 as complement for Table. 5. Our $S_{preserve}$ and $S_{modify}$ consistently yield better alignment towards preservation and modification compared to L2 and CLIPScore for each pair of editing model comparison.

### B.7. User Study Results for Masking Guidance Editing Enhancement

The detail for this user study can be found in Sec. A.3. As shown in Table. 15, the results of user study indicated that for all evaluation dimensions, the number of mask-guided models that surpassed original models was significantly higher than those that performed worse. Notably, mask guidance provided a remarkable improvement in maintaining regions outside the editing area, with 70% of the samples performing better than the original models.

## C. Qualitative results

### C.1. Visualization of Mask Quality Comparison

In Fig. 12, we compare our mask with GPT-4o generated bounding box and DiffEdit(Couairon et al., 2022) generated mask, our mask and bounding box can more accurately locate the editing region, providing more precise separation for evaluation, thoroughly yielding more trustworthy evaluation score.

**Editing instruction:**

turn the tie into a bow tie

Figure 11: **Visualized of human score collection procedure.** By offering references for identical input (Generated image 1-4 in each line), the annotators can judge the editing quality more reasonably. On the contrary, cross-sampled score collection lacks absolute comparison. For instance, scoring samples for two distinct instructions in column 3 (Generated image 2) is difficult to determine whether the "tie change case"(in first line) or "television change case"(in second line) is better in editing quality.

## C.2. Visualization of BPM evaluation

More qualitative results of BPM evaluation is shown in Fig. 13. For each input image and instruction along with their different editing results, our BPM can accurately verify the editing images' quality from different perspectives. For example, in Fig. 13(a), edited image 4 owns totally clean plate, which accurately follows the instruction "mask the plate empty", and the other region remains perfectly unchanged. BPM rate it with highest score among four editing images, validating the authenticity of BPM serving as evaluation metric.

## C.3. Visualization for mask-guided enhancement

Visualization of mask guided enhancement in shown in Figure. 14. Our mask accurately locate the region that should be edited, and successfully maintain the content of irrelevant regions, yielding more satisfactory editing result.

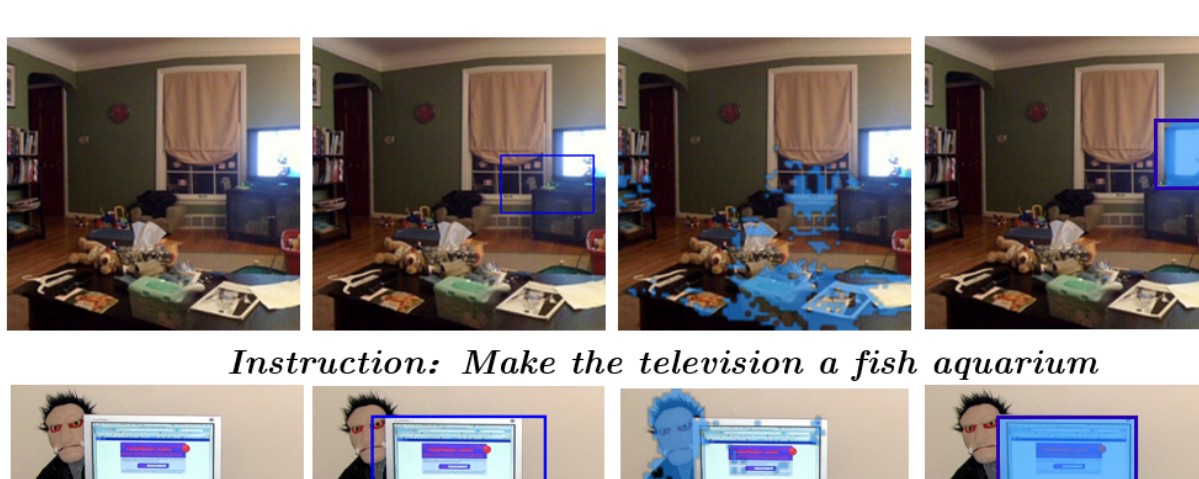

Instruction: Make the television a fish aquarium

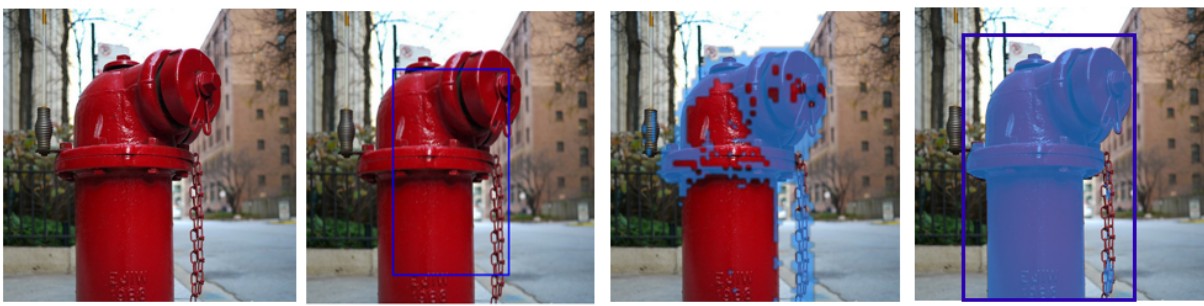

Instruction: Turn off the computer

Instruction: Make the hydrant blue

Original Image     GPT-4o     DiffEdit     BPM(Ours)

Figure 12: **Visualization of mask comparison.** We compare our editing region mask with two approaches, GPT-4o(OpenAI, 2024) and DiffEdit(Couairon et al., 2022). For GPT-4o, we prompt it to output the bounding box coordinates of editing region; for DiffEdit, we visualize the attention map that serves as their mask guidance during editing process.

Table 11: **The influence of the choice of LLM.** Experiment conducted on a 100-entries subset.

|  | mgie vs ft_ip2p | mgie vs ip2p | ft_ip2p vs ip2p | dalle2 vs ip2p | dalle2 vs ft_ip2p | dalle2 vs mgie | Average |
|---|---|---|---|---|---|---|---|
| gemma | **0.761** | **0.700** | 0.831 | 0.866 | **0.823** | 0.773 | 0.793 |
| gpt-4o | **0.761** | 0.689 | **0.854** | 0.876 | 0.792 | **0.804** | 0.796 |
| gemini | 0.739 | 0.689 | **0.854** | 0.866 | 0.792 | 0.784 | 0.789 |
| deepseek-v3 | 0.739 | 0.689 | **0.854** | **0.887** | 0.823 | 0.804 | **0.799** |

Table 12: **The accuracy of instruction parsing.**

|  | Object identify | Size | Position |
|---|---|---|---|
| Accuracy | 0.98 | 0.99 | 0.97 |

Table 13: **Human ratings of mask generated by different methods.**

|  | GPT-4o | DiffEdit | Ours |
|---|---|---|---|
| Human rating | 2.66 | 1.45 | **4.01** |

Table 14: **Effectiveness verification for component score of BPM .** The component scores of BPM targets for different perspective evaluation of editing quality, we compare them with corresponding human evaluation(specific scores in the target perspective).

|  |  | MGIE vs FT$_{IP2P}$ | MGIE vs IP2P | IP2P vs FT$_{IP2P}$ | DaLLE-2 vs IP2P | DALLE-2 vs FT$_{IP2P}$ | DALLE-2 vs MGIE |
|---|---|---|---|---|---|---|---|
| preservation | L2 | **0.756** | 0.816 | 0.816 | 0.893 | 0.743 | 0.927 |
|  | $S_{preserve}$ | 0.729 | **0.839** | **0.829** | **0.893** | **0.757** | **0.927** |
| modification | CLIPScore | 0.617 | 0.645 | 0.624 | 0.484 | 0.479 | 0.515 |
|  | $S_{modify}$ | **0.678** | **0.686** | **0.659** | **0.67** | **0.606** | **0.583** |

Table 15: **User Study for our Mask Guidance Enhancement.** We randomly select 100 image samples for comparison. "Ours" denotes edited image under our mask guidance is preferred by user, "Equal" denotes no obvious difference between original and our edited image, "Origin" denotes original edited image is preferred.

|  | Ours | Equal | Origin |
|---|---|---|---|
| Position | 38 | 54 | 8 |
| Size | 23 | 75 | 2 |
| Modification | 38 | 48 | 14 |
| Preservation | 70 | 26 | 4 |
| Overall | 57 | 34 | 9 |

| Original Image | Edited Image 1 | Edited Image 2 | Edited Image 3 | Edited Image 4 |
|---|---|---|---|---|
| 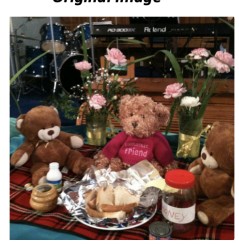 | 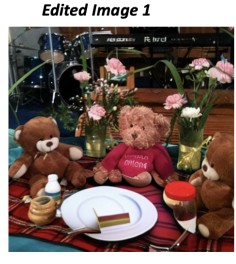 | 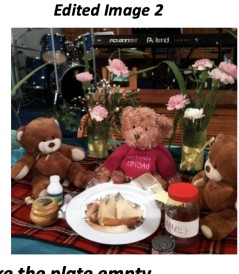 | 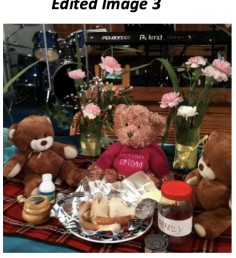 | 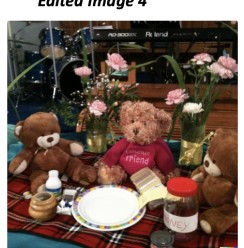 |

*Instruction: make the plate empty .*

| | | | | |
|---|---|---|---|---|
| LPIPS↓ | 0.158 | 0.102 | **0.063** | 0.086 |
| L2↓ | 0.093 | 0.054 | 0.051 | **0.047** |
| CLIPScore↑ | **0.180** | 0.159 | 0.159 | 0.144 |
| GPT-4o↑ | 78.5 | 40.0 | 47.5 | **84.0** |
| BlanceCLIP-$S_{modify}$↑ | 0.128 | 0.110 | 0.071 | **0.283** |
| BlanceCLIP-$S_{preserve}$↑ | 0.865 | 0.915 | 0.914 | **0.964** |
| BlanceCLIP-$S_{semantic}$↑ | 0.993 | 1.025 | 0.985 | **1.248** |
| BlanceCLIP-$S_{size}$↑ | 1 | 0 | 0 | 1 |
| BlanceCLIP-$S_{position}$↑ | 1 | 0 | 0 | 1 |
| BlanceCLIP-$S_{overall}$↑ | 2.993 | 1.025 | 0.985 | **3.248** |

(a)

| Original Image | Edited Image 1 | Edited Image 2 | Edited Image 3 | Edited Image 4 |
|---|---|---|---|---|
| 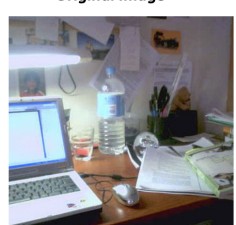 | 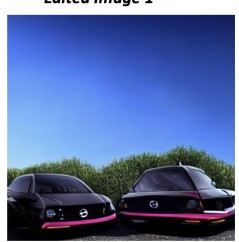 | 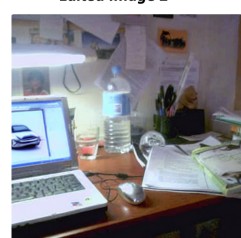 | 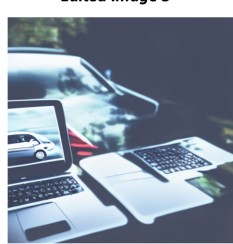 | 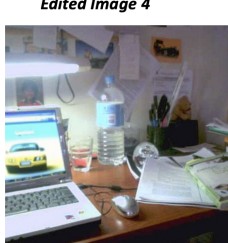 |

*Instruction: Add a car on the screen.*

| | | | | |
|---|---|---|---|---|
| LPIPS↓ | 0.687 | **0.051** | 0.534 | 0.052 |
| L2↓ | 0.283 | 0.003 | 0.215 | **0.003** |
| CLIPScore↑ | **0.189** | 0.167 | 0.187 | 0.167 |
| GPT-4o↑ | 25.5 | 77.5 | 56.5 | **85** |
| BlanceCLIP-$S_{modify}$↑ | 0.274 | 0.391 | 0.264 | **0.525** |
| BlanceCLIP-$S_{preserve}$↑ | 0.342 | 0.952 | 0.482 | **0.973** |
| BlanceCLIP-$S_{semantic}$↑ | 0.617 | 1.343 | 0.746 | **1.499** |
| BlanceCLIP-$S_{size}$↑ | 1 | 1 | 1 | 1 |
| BlanceCLIP-$S_{position}$↑ | 0 | 1 | 1 | 1 |
| BlanceCLIP-$S_{overall}$↑ | 1.617 | 3.343 | 2.746 | **3.499** |

(b)

Figure 13: **Visualized Evaluation comparison among different metrics.**

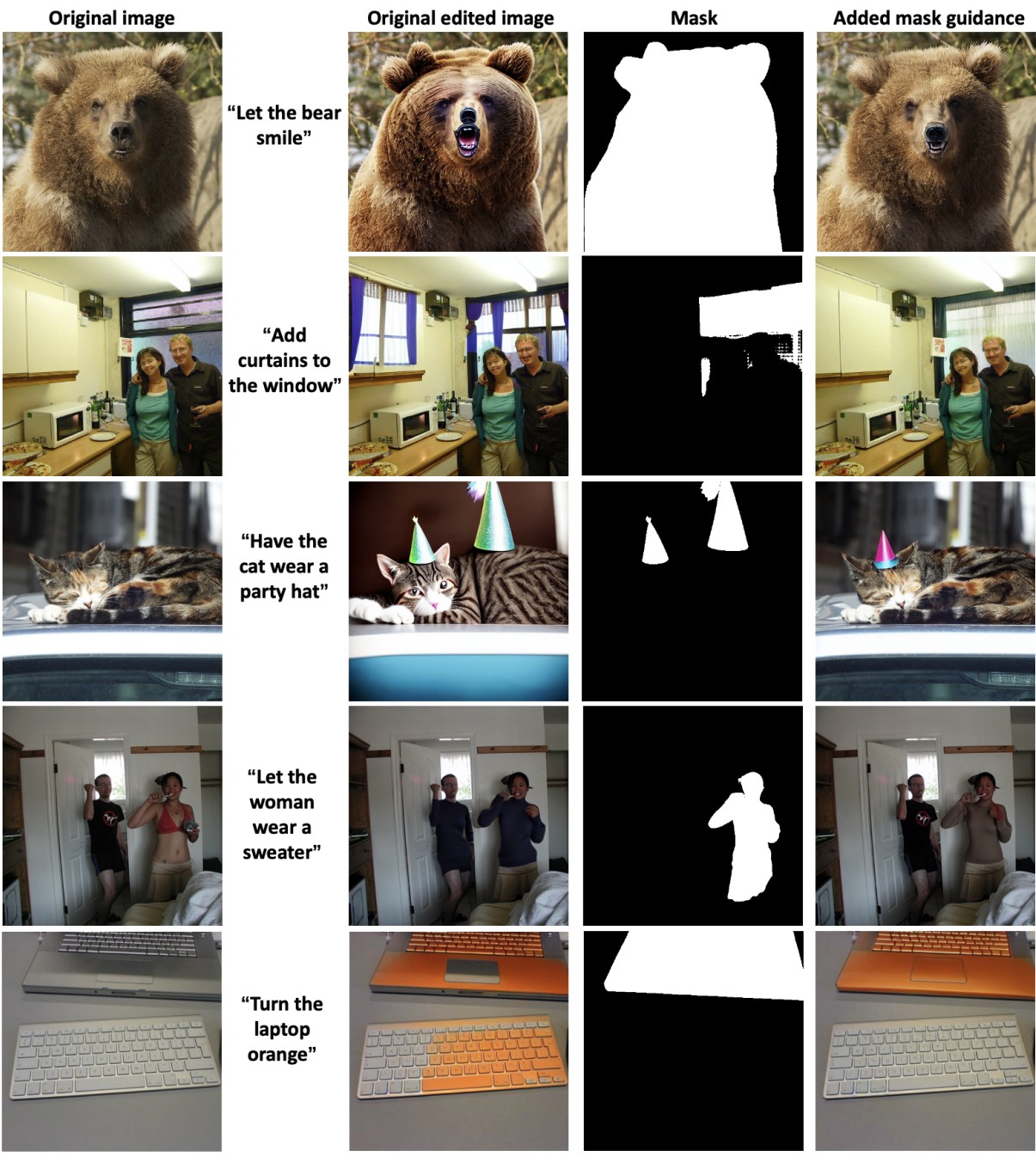

Figure 14: **More visualization of mask-guided editing.**

