# OpenReview forum: "Balancing Preservation and Modification: A Region and Semantic Aware Metric for Instruction-Based Image Editing"
_ICML.cc/2025/Conference — ICML 2025 poster_

### Official Review · Reviewer_PHma · 2025-03-03

**Overall Recommendation:** 3

**Summary:**

This paper presents BPM, a new image quality assessment metric designed to address the biases of existing metrics in image editing, particularly regarding the preservation and modification areas, as well as their narrow considerations. BPM introduces a region-based approach by using masks to divide the image into preservation and modification areas, evaluating them separately across multiple dimensions, including position, size, preservation, and modification. Specifically, the calculation process of the metric minimizes the involvement of models, reducing biases introduced by the priors of other models. Experimental results show that, compared to existing metrics such as CLIPScore and LPIPS, BPM is closer to human subjective evaluations. Furthermore, incorporating BPM into model training can effectively improve the accuracy of edits.

**Claims And Evidence:**

The claims made in paper provides a thorough analysis of the issues with existing indicators, and the proposed viewpoints are supported by experimental evidence. BPM effectively maintains a balance in evaluation between the preservation and modification areas. However, the sample size used in the experiments is only 100, which lacks sufficient persuasive power.

**Essential References Not Discussed:**

[1] Han Z, Jiang Z, Pan Y, et al. ACE: All-round Creator and Editor Following Instructions via Diffusion Transformer[J]. arXiv preprint arXiv:2410.00086, 2024.

[2] Wang Z, Li A, Li Z, et al. Genartist: Multimodal llm as an agent for unified image generation and editing[J]. Advances in Neural Information Processing Systems, 2025, 37: 128374-128395.

**Experimental Designs Or Analyses:**

The organization of the experiments is reasonable, with clear logical progression and evidence between them. The comparative and ablation experiments successfully demonstrate the advantages of BPM over other metrics. However, the limited experimental evaluation metrics in Section 4.4 fail to convincingly support the claims made in Section 3.5. In addition, the relatively small dataset in other experiments makes it difficult to provide strong persuasive power.

**Methods And Evaluation Criteria:**

The BPM proposed in the paper draws on the region-based concept while considering the impact of instructions on the preservation and modification areas, which is reasonable for the current problem. The diverse evaluation metrics compared on dataset A are robust, as the reference metrics include a wide range of multi-category evaluation indicators, such as CLIP-Score, GPT-4o, and LPIPS.

**Other Comments Or Suggestions:**

If the paper could conduct experiments on a larger dataset, its persuasive power would be stronger.

**Other Strengths And Weaknesses:**

Pros:

1.	BPM excels in maintaining and modifying the balance of regions compared to other existing metrics, and the results in Table 2 of the paper reveal the underlying reasons.

2.	BPM, when applied to model optimization, demonstrates more precise editing effects.

3.	The impact of multi-class LLM priors on BPM is discussed, and the results in Table 8 indicate that the BPM calculation process effectively reduces the influence of LLM priors.

4.	The writing and experimental organization are clear and make complex concepts understandable.

Cons:

1.	The dataset used in the experiment is relatively small, making it difficult to provide sufficient persuasive power.

2.	The scoring design for position struggles to handle complex orientations.

**Questions For Authors:**

1.	Could the authors provide experimental results on a larger dataset?

2.	Could the authors further elaborate on the design rationale behind the position and size evaluation process?

3.	Could the authors provide more experimental results in Section 4.4, including the performance of the edited images generated by the BPM-trained model on other metrics?

**Relation To Broader Scientific Literature:**

The paper builds the metric design based on the existing popular region-based ideas in the field of image editing.

**Theoretical Claims:**

The theoretical analysis is relatively detailed, but the scoring design for position and size evaluation is unreasonable. There is no further in-depth explanation of the rationality behind the design of the metrics.

---

> ### Author Rebuttal · Authors · 2025-04-01
>
> We appreciate the reviewer's recognition of our work that "viewpoints are supported by experimental evidence", "theoretical analysis is detailed", "experiments is reasonable with clear logical progression and evidence"
>
> **1.Verification on larger dataset:**(1) As suggested, we conduct experiments on a larger dataset. We curate a dataset with a total of 1000 samples, which is currently the largest for image editing metric evaluation. Images are randomly selected from MagicBrush and edited by 6 advanced editing models, which include the reviewer suggested ACE and Genartist model.
> The results on the newly expanded dataset is shown in Table1. Our BPM yield highest human alignment score on this dataset as well, validating its effectiveness and robustness.(2)Although our BPM metric calculation is fully automated, the user study required for human alignment test hinders further dataset scale-up.
> |dataset |MGIE vs FT_IP2P $\uparrow$| MGIE vs IP2P$\uparrow$ | IP2P vs FT_IP2P $\uparrow$ |DALLE-2 vs IP2P $\uparrow$  |FT_IP2P vs DALLE-2 $\uparrow$| DALLE-2 vs MGIE $\uparrow$ | ACE vs GenArtist $\uparrow$ |Average|
> |---|---|---|---|---|---|---|---|---|
> CLIP-I | 0.59 | 0.581 | 0.58 | 0.708 | 0.663 | 0.766 | 0.757 | 0.663|
> CLIP-T | 0.538 | 0.477 | 0.519 | 0.375 | 0.407 | 0.34 | 0.608 | 0.466|
> LPIPS | 0.677 | 0.605 | 0.691 | 0.711 | 0.628 | 0.766 | 0.838 | 0.709 |
> DINOScore | 0.59 | 0.581 | 0.63 | 0.781 | 0.733 | 0.798 | 0.784 | 0.699|
> L2 | **0.705** | 0.628 | 0.691 | 0.698 | 0.605 | 0.734 | 0.824 | 0.697 |
> GPT-4o |  0.667 | **0.674** |0.790 | 0.906 | 0.779 | 0.83 | 0.703 | 0.764|
> BPM(Ours) | 0.679 | 0.616 | **0.827** | **0.938** | **0.814** | **0.904** | **0.824** | **0.800** |
> **Table 1**
>
> **2.Rationality behind $f_{position}$ and $f_{size}$:**(1) Our intension is that previous metrics evaluate the entire image, thereby overlooking region-specific semantic requirements and failing to assess whether the fine-grained state changes of edited objects align with instructions. Size and position serve as typical state changes, thus we design corresponding function for their evaluation. (2) regarding how to evaluate the size and position change to make it more generalisable,  we also compare GPT-4o that directly judge whether the editing obeying corresponding state change demanding. The results are shown in  Table 2, our tailored functions $f_{size}$ and $f_{position}$ yield more human-aligned judgment, validating the trustworthiness.
> |dataset |MGIE| IP2P | DALLE-2|FT_IP2P|Average|
> |---|---|---|---|---|---|
> |$S_{size}$| | | | | |
> |GPT-4o | **0.92** |**0.90** | 0.93 | 0.90 |0.91|
> |BPM(Ours)| 0.89 | 0.89 | **0.93** | **0.92** |**0.91**|
> |$S_{position}$| | | | | |
> |GPT-4o | **0.71** | 0.69 | 0.89 | 0.75 |0.76 |
> |BPM(Ours)| 0.69 | **0.75** | **0.92** | **0.90** |**0.82**|
> **Table 2**
>
> **3.More experimental results on BPM-based editing images:** Apart from BPM evaluation, we further provide following results:
> (1)_User study:_ we conduct user preference tests, which is provided in Table12 of Supp.M. BPM-based editing yield 91\% of the samples being rated as better or equally
> good as the original compared to original ones, validating BPM serves as effective editing quality enhancement.
> (2) _Evaluation with other metrics:_ We also evaluate images by other metrics, which is shown in Table3. As shown, our BPM-based editing images yield the better performance for all these metrics as well, validating the robustness of BPM quality enhancement.
> |Metrics |CLIP-I $\uparrow$|LPIPS$\downarrow$ | DINOScore$\downarrow$ | L2$\downarrow$|CLIP-T$\uparrow$|GPT-4o$\uparrow$|
> |---|---|---|---|---|---|---|
> | IP2P  | 0.773 |0.315 |0.854 | 0.147 | 0.173 |0.559|
> |IP2P+BPM(Ours) | **0.923** | **0.103** | **0.997**|**0.047**|**0.176**|**0.667**|
> FP_IP2P |0.870 | 0.160|0.945 | 0.071 | 0.175 | 0.673 |
> FP_IP2P + BPM(Ours) | **0.911**|**0.135** | **0.951**|**0.063**| **0.175**| **0.720**|
> MGIE | 0.856 | 0.218 | 0.921 | 0.102 |0.176 | 0.613 |
> MGIE+BPM(Ours)|**0.895**|**0.124**|**0.950**|**0.065**|**0.177**|**0.701**|
> **Table 3**
>
> **4.Inclusion with more related works:**(1)Suggested works ACE and GenArtist focus on new  framework for image generation and editing, but they all use conventional metrics for evaluation. In contrast, our main contribution is the new evaluation metrics.
> (2)Experimental comparisons of testing them using our new metrics are included in  Table 1, and we will add relative discussion in related work in final version.

---

### Official Review · Reviewer_kMZ5 · 2025-03-08

**Overall Recommendation:** 3

**Summary:**

This paper proposes a novel evaluation metric for instruction-based image editing. The paper identifies the weaknesses in commonly used metrics, which can only consider the image as a whole but fail to focus on local and modified parts. It proposes to extract the edited object and intended editing effect from the editing instruction using VLLM, and use a detector and segmentor to locate these objects from original and edited images. It then proposes to decompose each perspective of the editing and compare the position, shape, and semantics of the original and edited objects separately to determine the alignment of the editing to the instruction. The paper conducts an experiment on the MagicBrush benchmark and demonstrates that the proposed evaluation metric is more aligned to the human evaluation. It also applies the object mask extracted by the segmentor to the diffusion process, improving the overall editing result.

# Update after rebuttal
I am grateful to the author's clarification on the proposed metric working under global editing with experimental results. I also appreciate that authors have taken my concern regarding the Human Alignment Test into consideration and performed additional experiments on human evaluation correlation, and promise to include these to the final version. My concerns are largely solved. Therefore, I raised my rating to "weak accept".

**Claims And Evidence:**

The first claim is the identification of the weaknesses of commonly adopted metrics, such as CLIPSCore, L2, etc. The analysis about this part is solid because CLIP or LPIPS indeed consider the image as a whole and cannot focus on one specific region. Therefore, this claim is solid. The ground truth test in Table 2 further substantiates this point.

The second claim is the proposed metric. It uses LLM to extract the original and edited objects and compare the position, size, and semantics of them separately, and it indeed addresses the problem of most previous metrics that consider the entire image as a whole. This is a solid contribution. However, the scope of the proposed metric is worth more discussion, and I will talk about this in the **Methods And Evaluation Criteria** section of the review. Moreover, In **Semantic Instruction Compliance**, the metric for measuring modified region $S_{modify}$ is the same as the CLIP directional similarity [1]. There is no reference to it, and $S_{modify}$  is claimed as a contribution in Ln432 and Table 5. This is an overclaim. Despite this, the rest of the metric is novel.

The effectiveness of the proposed metric is shown by Tables 1, 2, 3, and 4. They show that the proposed metric aligns better to human evaluation over the baseline metrics such as CLIPScore or LPIPS. However, its benefit over its VLLM competitor HQ-Edit, labeled as GPT-4o in the main paper, is worth more discussion, which I will talk about in detail in the **Experimental Designs Or Analyses**.

The third claim is that inserting the object mask extracted by the segmentor to the classifier-free guidance of the diffusion process. The idea of locating to-be-edited objects to alleviate undesired effects is not new. Works like Prompt2prompt [2] and DiffEdit [3] use the attention map in the diffusion model to locate to-be-edited objects. In this work, object masks are obtained using an external segmentor, following the same path. The effect is demonstrated in Table 6.


[1] Kim, Gwanghyun, Taesung Kwon, and Jong Chul Ye. "Diffusionclip: Text-guided diffusion models for robust image manipulation." In Proceedings of the IEEE/CVF conference on computer vision and pattern recognition, pp. 2426-2435. 2022.

[2] Hertz, Amir, Ron Mokady, Jay Tenenbaum, Kfir Aberman, Yael Pritch, and Daniel Cohen-Or. "Prompt-to-prompt image editing with cross attention control." arXiv preprint arXiv:2208.01626 (2022).

[3] Couairon, Guillaume, Jakob Verbeek, Holger Schwenk, and Matthieu Cord. "Diffedit: Diffusion-based semantic image editing with mask guidance." arXiv preprint arXiv:2210.11427 (2022).

**Essential References Not Discussed:**

The literature review contains all essential works to understand the key contributions of the paper.

**Experimental Designs Or Analyses:**

As discussed in the **Method and Evaluation Criteria** section, the benchmark only consists of local object-oriented editing without global editing. I am worried that this may unfairly benefit the proposed metric over other metrics, especially over the GPT-4o baseline, in the experiment because the proposed metric is specifically designed for local editing. It competes against other metrics in its strong suit. Experiments on global editing are required to establish a clearer insight into the proposed metric.

Regarding the Human Alignment Test, it reports a metric's alignment to the human evaluation on choosing the better one from two given models. Why choose this format over the traditional correlation between metric score and human evaluation score on one single model? What are the benefits of the current format?

The experiment part fails to mention the details of the detector and segmentor used in the proposed method.

**Methods And Evaluation Criteria:**

My primary concern about the proposed metric is its limited scope. I recognize that the proposed metric is particularly suitable for local object-oriented editing such as adding, removing, transforming, and removing objects. However, the proposed method may not address the editing on the global scale, such as changing the style of the image or environment of the scene. Throughout the paper, global editing is not mentioned, and it is not in the benchmark used in the experiment. Its main VLLM-based competitors, Aligmence and Coherence in HQ-Edit, instead can address the evaluation of global editing but also have a decent performance in local editing at the same time, as shown in Table 1, where GPT-4o (HQ-Edit) ranks at the second place.

Moreover, even if we limit the scope to local object-oriented editing, my second concern is about the editing involving changing the object's color or texture rather than the object's category or size. According to Figure 6 Object identify, the LLM only extracts the object category from the instruction. If I have an instruction, "Change the color of the mug to blue", the original image and edited image will both be "mug", and the color change will not be reflected by $S_{modify}$ because $o_{1}$ and $o_{2}$ will be the same. In addition, CLIP is known to be good at identifying object category but not color or texture. This further limits the scope of the proposed evaluation metric.

**Other Comments Or Suggestions:**

There are several non-technical errors in the paper:

1. There are repetitive sentences at the end of the literature review **Conventional Image Editing Quality Evaluation Metrics**. Sentences between Ln116-Ln122 and sentences between Ln123-Ln129 talk about the same thing.

2. The sentence between Ln142-Ln146 is confusing with grammar mistakes.

3. The citation to MGIE is incomplete. The paper title and publication details are missing.

**Other Strengths And Weaknesses:**

All strengths and weaknesses are discussed in the previous sections.

**Questions For Authors:**

I will consider upgrading my rating if my concerns can be addressed. They are summarized as:

1. The limited scope of the proposed metric.
2. Missing experiments on global editing.
3. Benefits of the Human Alignment Test on two models over correlation to human evaluation score on one single model.
4. Details on the detector and segmentor.
5. The corner case on $f_{size}$ and $f_{position}$.

**Relation To Broader Scientific Literature:**

The paper concerns the evaluation of instruction-based image editing, which is a longstanding challenge in generative AI. The idea of this work comes from the analysis of the prevalent metrics, realizing that they fail to focus on the local part in editing.

**Theoretical Claims:**

This paper is an application paper with little theoretical contribution. However, I do notice an error in Equation 2, which is the standard classifier-free guidance formulation for image editing proposed in instructpix2pix [1]. The third line of the equation should be $s_{T} * (\epsilon_{\theta}(z_{t}, t, I_{origin}, T_{edit})-\epsilon_{\theta}(z_{t}, t, I_{origin}, \varnothing)*M_{edit}$. I believe it is due to a typo. However, if it is not a typo, justification needs to be shown for the change.

---

> ### Author Rebuttal · Authors · 2025-04-01
>
> We appreciate the reviewer's recognition of our work that "claim is solid", "metric is novel".
>
> **Limited scope of proposed metric, i.e., global editing and object attribute changing:** 1) Thanks for recognize our proposed metric suitable for local object-oriented editing. 2) As suggested, we apply our BPM to **global instruction-based editing** including style change, environmental change. a) *visualized results* is in [Fig](https://anonymous.4open.science/r/r-2/4.pdf). b)*Quantitative verification* we use the 160 global editing samples in PIE-bench for human alignment evaluation. Results are in [Tab](https://anonymous.4open.science/r/r-2/5.pdf). BPM yield highest human alignment for such global editing cases. c) *Comparison with GPT-4o(VLLM competitors):* While applicable, GPT-4o(HQ-Edit) underperforms our BPM in global editing. d) Unlike metrics like FID(conventionally used for style transfer) that rely on paired GT before-and-after images (“after” indicates target stye image), our proposed metric operates without this constraint, offering greater practicality, our metric operates without this constraint, offering greater practicality.3) Regarding **attribute changing**(a)*Clarify object identify step* for attribute changing cases, source/target object $o_1,o_2$ would affiliate corresponding attributes, like, "yellow mug" as $o_1$, "blue mug" as $o_2$ for "turning yellow mug into blue" b) *Attribute understanding* Our contribution is decompose image into editing-relevant and irrelevant regions for separate evaluation, allowing any CLIP variant to be applied. In our rebuttal, we test Long-CLIP-specialized in fine-grained attribute understanding-on a subset of 80 attribute-changing samples ([Tab](https://anonymous.4open.science/r/r-2/6.png)). Results show BPM’s evaluation capability for such cases can be improved with more advanced image encoder.
>
> **Benefits of Human Alignment Test on two models comparison rather than single model:** 1) As suggested, we report metric and human score correlation for each single model in [Tab](https://anonymous.4open.science/r/r-2/2.pdf). Though our BPM yields highest correlation in this format as well, such calculation is questionable as output from distinct instruction lacks comparability, and all metrics show performance not high.2)The rationale behind our pair model preference calculation is a) *Reduce Subjectivity influence*: Scores among users exhibit variance of 1.06(out of 0-5 range), thus using absolute values for single model correlation calculation may introduce bias. In contrast, employing relative preferences, by introducing a reference, reduces subjectivity in user study to enhance the credibility of the results. b) *Reduce cross-sample bias*: our metric evaluates which model performs better for the same instruction, rather than comparing scores across different instructions (as we believe outputs from distinct instructions lack absolute comparability). By pair-wise comparison on identical instructions, we enable more accurate model-to-model comparisons explicitly than correlating scores from a single model's diverse outputs.
>
> **Details of the detector and segmentor** We use Grounding-DINO 1.5 and SAM-large as our detector and segmentor respectively. Will add this in final version.
>
> **Corner case of $f_{position}$ and $f_{size}$** In [Fig](https://anonymous.4open.science/r/r-2/1.png), for object addition type instruction like "add dogs to the right of the surfboard",1) regarding $f_{size}$ (Line 8 of Alg.1, main text), since original object $o_1$ is None, $S_{size}$ is set as 1.2) regarding $f_{position}$ (Line 7 of Alg.1, main text), since $pos_{st}$ is "right", LLM would replace $o_1$ by the reference object surfboard, and judge whether $o_2$ dog obey "right of" $o_1$ surfboard.
>
> **Fairness of visualized comparison with other metrics:** (1)We provide more visualizations about the metric human alignment comparison [Fig](https://anonymous.4open.science/r/r-2/7.pdf). Across diverse types of editing instructions involving local attribute editing, object addition, global style transfer, our BPM can yield higher human alignment compared to GPT-4o and other metrics.
> (2)this mask comparison experiment in Fig 11.Supp only aims to justify the design choice of proposed route to locate mask.
>
> **Apply mask into diffusion to for edit is not new.** Agreed, 1)Our main contribution is new image editing metric that includes a novel pipeline for precise edited-region mask acquisition.2)this application is not our core contribution. It aims to show BPM design principle has broader potential as a byproduct for image editing.
>
> **Add reference to CLIP directional similarity:** Agreed, we will add reference and clarify in related work, fix overclaim point in final version. Thanks for recognize the rest of the proposed metric is novel.
>
> **Typo in Eq.2:** Thanks. We will correct this typo and incorporate all other valuable non-technical writing suggestions in the final version.

---

> > ### Comment · Reviewer_kMZ5 · 2025-04-04
> >
> > Thank you for your detailed response, which addressed most of my concerns. However, I still have two questions. 1) Based on the response, does it mean that the dominant metric would be S_modify as the rest scores are approaching zero in the case of global editing? 2) I do acknowledge the author's arguments regarding the Human Alignment Test. However, I still believe the correlation between human evaluation and metric score is better. What if someone wishes to benchmark a new model against all existing models? They have to repeat the Human Evaluation Test for every single existing model, which is very burdensome. The large variance of human evaluation can be reduced by expanding the number of samples. On the other side, I do acknowledge the cost of organizing a large-scale human evaluation, and it is not expected for rebuttal.

---

> > > ### Author Response · Authors · 2025-04-07
> > >
> > > Thanks for recognizing we have addressed most of the concerns, we respond the remaining questions below:
> > >
> > > **1.For global editing case, is $S_{modify}$ a dominant metric?**(1)Regarding Region-aware Judge $S_{region}$ of our BPM,for global editing,no position/size change is needed, so we set $S_{position}$,$S_{size}$ to 1(instead of 0) to indicate no such error occur(2)Regarding Semantic-aware Judge $S_{semantic}$,$S_{modify}$ is not always dominant:(a)For global environment  and weather change cases,certain region require preservation([Fig](https://anonymous.4open.science/r/r-4/3.png)),thus $S_{preserve}$ plays critical evaluative role.Both $S_{modify}$,$S_{preserve}$ align with human preference and collectively contribute to overall BPM. b) For extreme cases style transfer, where nearly every pixel may be altered, editing region mask could be entire image,both semantic modify(style change) and preservation(content preserve) should be evaluated within the same mask ([Fig](https://anonymous.4open.science/r/r-4/4.png)).Both $S_{preserve}$,$S_{modify}$ contribute to Semantic-aware Judge,show trends consistent with human preference.We will incorporate the discussion in final version.(3)Note beyond global editing,our BPM is crucial for local editing evaluation.Region metrics(size/position),semantic metrics(change/preservation) are both crucial for comprehensive evaluation([Fig](https://anonymous.4open.science/r/r-4/5.png))
> > >
> > > **2.Correlation between human evaluation and metric score is better than human evaluation on choosing the better one from paired models**(1)We report in both evaluation protocols(correlation in [Tab](https://anonymous.4open.science/r/r-4/1.png),pairwise model preference in Tab.1main text),our BPM show highest correlation with human preference under both protocols.Notably,BPM runs 10× faster than second-best metric(GPT-4o) and avoid costly GPT-4o fees.We will add results in final version for comprehensive discussion(2)Motivation for pairwise human alignment evaluation is: a)for _human score collection in user study_,we follow conventional protocol in previous works MagicBrush,which offers different edited images for identical input image and instruction,and ask user to rate them in comparative manner,since comparing edited images for identical sample is more trustworthy than cross-sampled comparison([Fig](https://anonymous.4open.science/r/r-4/6.png)).b)for _human alignment test_,the comparable style of human score collection encourages us to mimic the procedure and conduct pairwise human alignment test among different editing models,which is naturally aligned with human score collection.c)To mitigate subjective bias in human score variance across different users(1.06 out of 0-5 score range),we formulate our evaluation criterion from strict numerical agreement to model rank alignment(comparing ranking orders derived from our metric versus human scores),better accounts for variability in human judgments while preserving evaluation robustness.
> > >
> > > **3. Benchmark new model against all existing models need re-evaluate on every single existing model**(1)We highlight _human alignment test_ is conducted to evaluate metric reliability by comparing their alignment with human scores, rather than evaluate quality of certain editing model. Our BPM achieves highest alignment with human,indicate it's a reliable alternative metric for human evaluation of editing quality(2)_If someone wish to benchmark a new editing model against all existing models_, they can adopt our BPM to report the score and directly compare with BPM score of other editing models,higher BPM indicate better quality. For this situation,human alignment test involves pairwise model comparison is unnecessary.(3)_If someone wish to benchmark a new editing metric against all existing metrics_, we provide sufficient diverse data with human scores(We will release collected human scores and source code of this work if paper accepted).They can follow our pair-wise model preference evaluation protocol to evaluate the reliability of new metric. For this situation,re-collect human score is not needed, they only need to calculate score using new metric,then use our collected human data to calculate alignment score.
> > >
> > > **4.The variance of human evaluation can be reduced by expanding the number of samples:**(1)We clarify the variance are calculated across different annotator for identical sample,rather than across different samples. So expanding sample scale might not be ideal for reducing such variance.(2)As suggested,we report performance on larger dataset([Tab](https://anonymous.4open.science/r/r-4/2.png)) to make results more convincing,which consists 1100 samples annotated by 3 users.a)The vairance is 1.07,which doesn't show obvious difference with previous variance(1.06).b)note for large-scale dataset results,same key conclusion stands: our BPM achieves the highest alignment with human scores,GPT-4o ranking second best,showing effectiveness of our BPM.

---

### Official Review · Reviewer_N886 · 2025-03-14

**Overall Recommendation:** 3

**Summary:**

This paper introduces BPM, a metric designed for instruction-based image editing. BPM explicitly separates images into editing-relevant and irrelevant regions, enabling more precise evaluation. It conducts a two-tier assessment from both regional and semantic perspectives to provide a comprehensive measure of editing quality. Additionally, BPM’s editing region localization can be integrated into image-editing models to enhance output quality, demonstrating its broad applicability.

## update after rebuttal
The authors’ rebuttal has clarified my previous concerns. Taking into account the other reviewers' feedback and the authors' response, I choose to maintain my original evaluation (Weak accept).

**Claims And Evidence:**

Yes.

**Essential References Not Discussed:**

None.

**Experimental Designs Or Analyses:**

I reviewed the experimental results presented in Tables 1, 2, 3, and 4, and they appear to be correct.

**Methods And Evaluation Criteria:**

Yes.

**Other Comments Or Suggestions:**

None.

**Other Strengths And Weaknesses:**

**Paper Strengths:**

The paper is well written. The main motivation is clear and easy to understand.

**Paper Weaknesses:**

1. The overall pipeline of BPM contains excessive text and needs to be more concise and clear. For example, the relationship between Object Identify, Size State, and Position State in Figure 2 is unclear. Additionally, the excessive lines and arrows make it difficult to grasp the entire pipeline.
2. The proposed BPM utilizes Region-Aware and Semantic-Aware Judges independently. Does this mean BPM operates more slowly? Have you tested the inference time of BPM and compared it with SOTA methods?
3. It appears that BPM-overall is obtained by summing $S_{semantic}$ and $S_{region}$. Have you explored assigning different weights to $S_{semantic}$ and $S_{region}$ to improve the assessment?

**Questions For Authors:**

See weakness.

**Relation To Broader Scientific Literature:**

This paper contributes to the broader scientific literature in several ways, particularly within the context of instruction-based image editing, semantic understanding, and region-specific modifications.

**Theoretical Claims:**

No, as there are no theoretical claims made.

---

> ### Author Rebuttal · Authors · 2025-04-01
>
> We appreciate the reviewer's recognition of our work that "enabling more precise evaluation", "broad applicability", "paper is well written", "motivation is clear and easy to understand" and "contributes to the broader scientific literature".
>
> **1.More concise figure of BPM pipeline:** Thanks for your valuable suggestions, according to which we provide a new version of Figure 2 in [Fig](https://anonymous.4open.science/r/r-2/3.png).
>
> **2.Inference speed analysis:** We provide inference speed comparison in Table7, Supp.Material..
> (1) Region-Aware and Semantic-Aware Judge are independent and can conduct in parallel, and will not affect inference speed.
> (2) our BPM has comparable inference speed ($\textless$1s per image) with other metrics like CLIP-I, and largely shrink the time cost compared with MLLM-based evluation, such as GPT-4o(11.4s per image), showing efficiency.
>
> **3.Assigning different weights to $S_{semantic}$ and $S_{region}$:** We conduct experiments with different weight factor $\alpha$ between $S_{semantic}$ and $S_{region}$, i.e., $BPM= \alpha*S_{semantic} + (1-\alpha)*S_{region}$, the results are shown in Table1.
> Among all combinations, $\alpha=0.7$ reaches the best human evaluation alignment, verifying that higher yet moderate semantic scale would yield better evaluation results.
>
>
> |$\alpha$ |0|0.1|0.2|0.3|0.4|0.5|0.6|0.7|0.8|0.9|1|
> |---|---|---|---|---|---|---|---|---|---|---|---|
> | Human ALignment Score| 0.302| 0.423| 0.598 | 0.659| 0.738| 0.792| 0.819|  **0.822**| 0.776| 0.769| 0.761|
> **Table 1**

---

### Official Review · Reviewer_xJQS · 2025-03-14

**Overall Recommendation:** 3

**Summary:**

The authors proposed a new metrics to balance the evaluation of preservation and modification for instruction-based image editing tasks. It is supposed to address some practical problems of image editing evaluation, by disentangling and measuring both the edited and untouched regions. The results show that the newly proposed metrics demonstrate a high correlation between human judgement and automatic numeric numbers. It is expected to have a higher impacts on image editing domains.

## update after rebuttal
I appreciate the efforts on designing the pipeline for evaluating the editing tasks automatically with a higher human alignment. The proposed method is a heuristic pipeline without learning, but it can serve as an intermediate solution which unblocks many emerging editing models, as an auxiliary metrics of user study. I also read the rebuttals and like the experiments on global editing tasks. Though not perfect and having some issues of complexity and internal errors, having it released in public will still help the community. So i will keep my original scores and encourage the authors to add these new results into the paper.

**Claims And Evidence:**

The claims in the paper are well evaluated with sufficient evidences.
For example, the newly proposed scores are shown to have a higher correlation with human preference.

**Essential References Not Discussed:**

The reference list is sufficient to me.

**Experimental Designs Or Analyses:**

The authors did extensive research on LLM selection and masking methods. The human alignment testing seems very effective.
Preservation seems easier to measure than modification since it is a simpler task and mask is supposed to largely improve the scores. Will it be better to apply a balanced factor between preservation and modification?

**Methods And Evaluation Criteria:**

The proposed method covers many editing cases especially object addition / replacement / removal, mostly object-oriented. So it moslty measure how the object position / size is changed. Then It leverages precomputed masks to segment edited regions / unedited regions and measure each of them, to check the preservation.

However, the metrics might not be robust for other types of editing including image enhancement, global image style transfer, soft editing like flare removal etc. Those non-mask based editing, or non-object involved editing may not be applied for measurement.

**Other Comments Or Suggestions:**

See above.

**Other Strengths And Weaknesses:**

See above.

**Questions For Authors:**

See above.

**Relation To Broader Scientific Literature:**

The manually-designed heuristic rules seem effective in short-term. But for a better impact, a multi-modal LLM might be more useful when we can distill the rules and the complicated pipelines involving multiple models into a unified model finetuned from a LLM. The current method has its limitations of intractable errors within the pipeline, and not very generalized to unseen tasks. Also it does not do any learning.

**Theoretical Claims:**

No theory in the paper.

---

> ### Author Rebuttal · Authors · 2025-04-01
>
> We appreciate the reviewer's recognition of our work that
> "expected to have a high impacts on image editing domains", "claims are well evaluated with sufficient evidences", "human alignment testing is effective".
>
> **1.Robustness of BPM on more diverse image editing tasks:** (1)Our BPM can apply to global image style transfer like weather change and environment change as well, we provide visualizations in [Fig](https://anonymous.4open.science/r/r-2/4.pdf). For quantitative evaluation, we conduct our experiments on the global editing subset of PIE-bench, which contains 160 samples. The results are shown in Table1. Our BPM yields the highest human alignment compared to other metrics, showing its effectiveness and wild application scope.
> (2) We want to clarify the focus of this paper is introducing new metrics for Instruction-Based Image Editing, specifically targeting edits that require clear semantic changes.  Suggested tasks like image enhancement or flare removal—which primarily improve image quality rather than alter semantics, fall outside the scope of our work.
> |dataset |PIEvs HQ_Edit $\uparrow$| PIE vs FT\_IP2P$\uparrow$ | HQ_Edit vs FT_IP2P $\uparrow$ |FT\_IP2P vs IP2P $\uparrow$  |IP2P vs HQ_Edit $\uparrow$|IP2P vs PIE $\uparrow$ |Average$\uparrow$|
> |---|---|---|---|---|---|---|---|
> CLIP-I | 0.966 | 0.861 | 0.417 | 0.6 | 0.52 | 0.85 | 0.702|
> CLIP-T | 0.897 | 0.833 | **0.917** | **0.667** | 0.76 | 0.825 | 0.817|
> LPIPS | **1.0** | **0.917** | 0.375 | 0.467 | 0.48 | 0.9 | 0.690  |
> DINOScore | 0.966 | 0.833 | 0.25 | 0.533 | 0.52 | 0.9 | 0.667 |
> L2 | 0.897| 0.889 | 0.333 | 0.4 | 0.52 | **0.925** | 0.661 |
> GPT-4o |  0.759 | 0.722 |0.583 | **0.733** | 0.76 | 0.825 | 0.730 |
> BPM(Ours) | 0.931 | **0.917** | 0.875 | 0.6  | **0.84** | 0.85 | **0.836** |
> **Table 1**
>
> **2.Apply balanced factor between preservation and modification:** Thanks for the suggestions. We conduct experiments with different balanced factor $\alpha$ between $S_{preserve}$ and $S_{modify}$, i.e., $S_{semantic} = \alpha*S_{preserve} + (1-\alpha)*S_{modify}$, the results are shown in Table2.
>  Among all combination, $\alpha=0.5$ reaches the best human evaluation alignment, verifying that moderate scale between preservation and modification yield better evaluation result.
>
> |$\alpha$ |0.1|0.3|0.5|0.7|0.9|
> |---|---|---|---|---|---|
> | Human ALignment Score| 0.647 |  0.704 |  **0.792**| 0.768 |0.732 |
> **Table 2**
>
>
> **3.Distill our rules in MLLM to evaluate in a unified model for better impact:** As suggested, we select most advanced closed-sourced MLLM GPT-4o and open-sourced MLLM qwen-vl-max-latest for our rule distillation. The results are shown in Table3, indicate
> (1) describe our design rules as prompt to guide GPT-4o via chain-of-thought ("GPT+rule", Line 2) improve human alignment of original GPT-4o(Line 1) by 2.7\%, showing our rules, that separately evaluate the preservation and modification with explicit region decomposition, are essential for comprehensive evaluation and can assist to yield more trustworthy results.
> (2) Though our evaluation rules can improve GPT-4o human alignment, it still falls short of our tailored pipeline(3.0\% performance drop, Line 2,5). This originates from that our pipeline endows LLM and segmentation tools for accurate editing region localization and explicitly evaluating preservation, modification separately, while MLLMs, even the most advanced GPT-4o, is sub-optimal for such localization task, as verified in our Fig.11,Supp. What' more, we further provide GPT-4o with our generated mask("GPT+rule+mask", Line 3) as condition for evaluation, but no significant performance improvement is observed, verifying that MLLM shows weaker evaluation interpretability than our pipeline and fail to fully utilize such accurate mask guidance.
> (3) Open-sourced MLLM Qwen(Line 4) lags behind in human alignment(12.4\% performance drop compared to BPM). Besides, there is no large-scale ideal data for such role distillation training. The data should contain paired real and edited images along with human-annotated editing masks,  and heavy human annotation on the quality judgment for modification, preservation, size and position change, so we don't do any learning like fine-tuning it. (4) Our rules witness consistent improvement in MLLMs evaluation, showing its wild application and potential for generalization towards more unseen tasks like more diverse image editing.
> |dataset |MGIE vs FT_IP2P $\uparrow$| MGIE vs IP2P$\uparrow$ | IP2P vs FT_IP2P $\uparrow$ |DALLE-2 vs IP2P $\uparrow$  |FT_IP2P vs DALLE-2 $\uparrow$| DALLE-2 vs MGIE $\uparrow$ | Average$\uparrow$|
> |---|---|---|---|---|---|---|---|
> GPT-4o | 0.693 | 0.678 | 0.64 | 0.814 | 0.781 | 0.804 | 0.735 |
> GPT-4o+rule | **0.784** | **0.744** | 0.73 | 0.876 | 0.625 | **0.814** | 0.762 |
> GPT-4o+rule+mask | 0.727|0.622|0.798|**0.918**|0.688|0.763|0.752|
> Qwen|0.614|0.633|0.73|0.804|0.562|0.67|0.668|
> BPM(Ours)|0.761|0.7|**0.831**|0.866|**0.823**|0.773|**0.792**|
> **Table 3**

---

### Decision · Program_Chairs · 2025-05-01

**Decision:**

Accept (poster)

**Comment:**

The final rating this paper is 4 weak accept.

Before rebuttal,

Reviewers acknowledged that 1) paper is well written and motivation is clear and easy to understand; 2) BPM excels in maintaining and modifying the balance of regions compared to other existing metrics. The main concerns from reviewers are: 1) the metrics might not be robust for other types of editing including image enhancement, global image style transfer, soft editing like flare removal etc; 2) The current method has its limitations of intractable errors within the pipeline, and not very generalized to unseen tasks; 3) proposed metric is its limited scope;

After rebuttal, all reviewers confirmed that they had read the author's response to the review and would change review if needed. Reviewer kMZ5 raised rating to "weak accept" and all other reviewers kept their rating. Reviewer xJQS mentioned "Though not perfect and having some issues of complexity and internal errors, having it released in public will still help the community. So i will keep my original scores and encourage the authors to add these new results into the paper." Reviewer N886 suggested "The authors’ rebuttal has clarified their previous concerns"

Based on these AC decided to give weak accept as final rating for this paper. But hope authors could incorporate reviewers' comments in their final version.